# Common mode signals and vertical velocities in the great Alpine area from GNSS data

Francesco Pintori[1], Enrico Serpelloni[1,2], Adriano Gualandi[1]

[1]Istituto Nazionale di Geofisica e Vulcanologia (INGV), Osservatorio Nazionale Terremoti, Roma, 00143, Italy.
[2]Istituto Nazionale di Geofisica e Vulcanologia (INGV), Bologna, 40128, Italy.

*Correspondence to*: Francesco Pintori (francesco.pintori@ingv.it)

**Abstract.** We study the time series of vertical ground displacements from continuous GNSS stations located in the European Alps. Our goal is to improve the accuracy and precision of vertical ground velocities and spatial gradients across an actively deforming orogen, investigating the spatial and temporal features of the displacements caused by non-tectonic geophysical processes. We apply a multivariate statistics-based blind source separation algorithm to both GNSS displacement time series and to ground displacements modeled from atmospheric and hydrological loading, as obtained from global reanalysis models. This allows us to show that the retrieved geodetic vertical deformation signals are influenced by environmental-related processes and to identify their spatial patterns. Atmospheric loading is the most important one, reaching amplitudes larger than 2 cm, but also hydrological loading, with amplitudes of about 1 cm, cause peculiar spatial features of GNSS ground displacements: while the displacements caused by atmospheric and hydrological loading are apparently spatially uniform, our statistical analysis shows the presence of NS and EW displacement gradients.

We filter out signals associated with non-tectonic deformation from the GNSS time series to study their impact on both the estimated noise and linear rates in the vertical direction. While the impact on rates appears rather limited, given also the long-time span of the time-series considered in this work, the uncertainties estimated from filtered time-series assuming a power law + white noise model are significantly reduced, with an important increase in white noise contributions to the total noise budget. Finally, we present the filtered velocity field and show how vertical ground velocity spatial gradients are positively correlated with topographic features of the Alps.

**Summary** We study time varying vertical deformation signals in the European Alps by analyzing GNSS position time series. We associate the deformation signals to geophysical forcing processes, finding that atmospheric and hydrological loading are by far the most important cause of seasonal displacements. Recognizing and filtering out non-tectonic signals allows us to improve the accuracy and precision of the vertical velocities.

## 1 Introduction

The increasing availability of GNSS observations, both from geophysical and non-geophysical networks, pushed forward the use of ground displacement measurements to study active geophysical processes on land, ice and on atmosphere, with applications in a broad range of Earth science disciplines (e.g., Blewitt et al., 2018). Studies on active mountain building, in particular, can now benefit from the use of GNSS vertical ground motion rates to get new insights into the contribution of the different processes at work in the formation and evolution of mountain reliefs (e.g., Faccenna et al., 2014a; Sternai et al., 2019, Dal Zilio et al. 2021, Ching et al. 2011). Proposed mechanisms of rock uplift rate include isostatic adjustment to deglaciation, tectonic shortening, isostatic response to erosion and sediment redistribution, isostatic response to lithospheric structural changes and dynamic adjustment due to sub-lithospheric mantle flow (e.g., Faccenna et al., 2014b). All these processes sum-up to contribute to the actual vertical ground motion rates estimated from GNSS displacement time-series, and constraining their relative contribution to mountain dynamics is challenging, because of the different spatial and temporal scales involved and the short observational time period with respect to the characteristic timescales of the mentioned processes.

The availability of long-lasting (i.e., >8 yrs) GNSS position time-series minimizes the impact of transient and seasonal signals in the vertical rate estimates (Masson et al., 2019). However, it is worth considering that GNSS measurements record ground displacements due to a variety of multiscale processes (from continental-scale geodynamics and loading to local-scale hydrology and tectonics), resulting in the presence of several deformation signals superimposed on the main linear trend, which is commonly associated with geodynamic processes at the scale of current, decadal, geodetic observation window.

Excluding tectonic and volcanological processes, and once removed the effect of tides associated with solid earth, pole and ocean, variations of atmospheric pressure loading and fluid redistribution in the Earth crust are the main cause of vertical ground displacement recorded by GNSS stations worldwide (Liu et al. 2015). Atmospheric pressure and mass changes cause time-variable displacement because of the elastic response of the Earth surface to these load variations, with vertical displacements usually significantly larger than the horizontal ones, which appear as spatially-correlated signals with a dominant one year period (e.g., Fu and Freymueller, 2012; Fu et al., 2012). Seasonal displacements are also caused by non-tidal sea surface fluctuations. This process is of particular relevance in areas near the oceans, while in the inlands its effect is significantly reduced (van Dam et al., 2012).

The presence of spatially-correlated signals in GNSS time-series can result from either the aforementioned large scale processes, generally described as common mode signals (CMS), or processing errors, generally described as common mode error (CME), like the mismodeling of displacements caused by solid Earth, ocean and atmospheric, and satellite orbits mismodeling, which induces draconitic signals (Dong et al., 2006).

In the literature, the distinction between CMS and CME is not always clear, and spatially correlated signals are often removed from the time series as CME without attempts of interpretation (e.g., He et al., 2017; Hou et al., 2019; Serpelloni et al., 2013; Kreemer and Blewitt, 2021). Depending on the pursued goal, this approach can be fair. For example, if we were interested in the study of long-term linear deformation, we might consider CMS as CME, but it is worth noting that the "CME" definition

for signals clearly associated with geophysical processes might be misleading. The removal of the CME/CMS in GNSS position time-series, which is also known as time-series filtering, can help improve the precisions of the estimated linear velocities. Moreover, a better understanding of CMS/CME origin can also provide new information on other deformation mechanisms.

Here we use the European Alps as a natural laboratory to investigate the spatial and temporal contribution of different geophysical processes, which we identify through a variational Bayesian Independent Component Analysis (vbICA), on the vertical ground displacements recorded by a dense and spatially uniform network of continuous GNSS stations in the 2010-2020 time-span. The Alps represent the highest and most extensive mountain range of Europe (see Fig. 1). We focus on the vertical component, which is nominally less accurate and precise than the horizontal ones, because this mountain belt is characterized by significant ground uplift and spatial vertical velocity gradients that are correlated with topography (Serpelloni et al., 2013). The present-day convergence between Adria and the Eurasian plate is largely accommodated in the Eastern Southern Alps (e.g., Serpelloni et al., 2016) where the Adriatic lithosphere underthrusts the Alpine mountain belt, and here part of the observed vertical uplift is associated with active tectonics (Anderlini et al., 2020). Conversely, in other Alpine domains, positive vertical velocities most likely derive from a complex interplay of deep-seated geodynamic and isostatic processes (e.g., Sternai et al., 2019). In the Alpine framework, more accurate and precise measurements of geodetic vertical ground motion rates can provide new constraints on the dynamics contributing to the ongoing vertical rates and their spatial variations, with implications for the study of mountain building processes, response to deglaciation and active tectonics.

The structure of this work is as follows: in Section 2 we present methods commonly used for extracting spatially-correlated signals in GNSS time series; in Section 3 we describe the data and methods used in this work; in Section 4 we characterize the spatio-temporal behavior of three different independent datasets (GNSS vertical displacements, atmospheric and hydrological loading models displacement time series) applying on each of them a vbICA decomposition and studying how they are related. This allows us to spatially and temporally characterize the signals contributing to the measured GNSS displacement time series and associate them with geophysical processes. We also estimate the vertical velocities and the noise features of the GNSS stations after removing the non-tectonic signals identified with the vbICA analysis. In Section 5 we compare the results of different filtering methods and use the results of our time-series analyses in order to evaluate the effects of the signal filtering on the accuracies and precisions of the vertical velocities of the study region, which is of particular importance to better characterize the processes generating the Alps uplift.

89

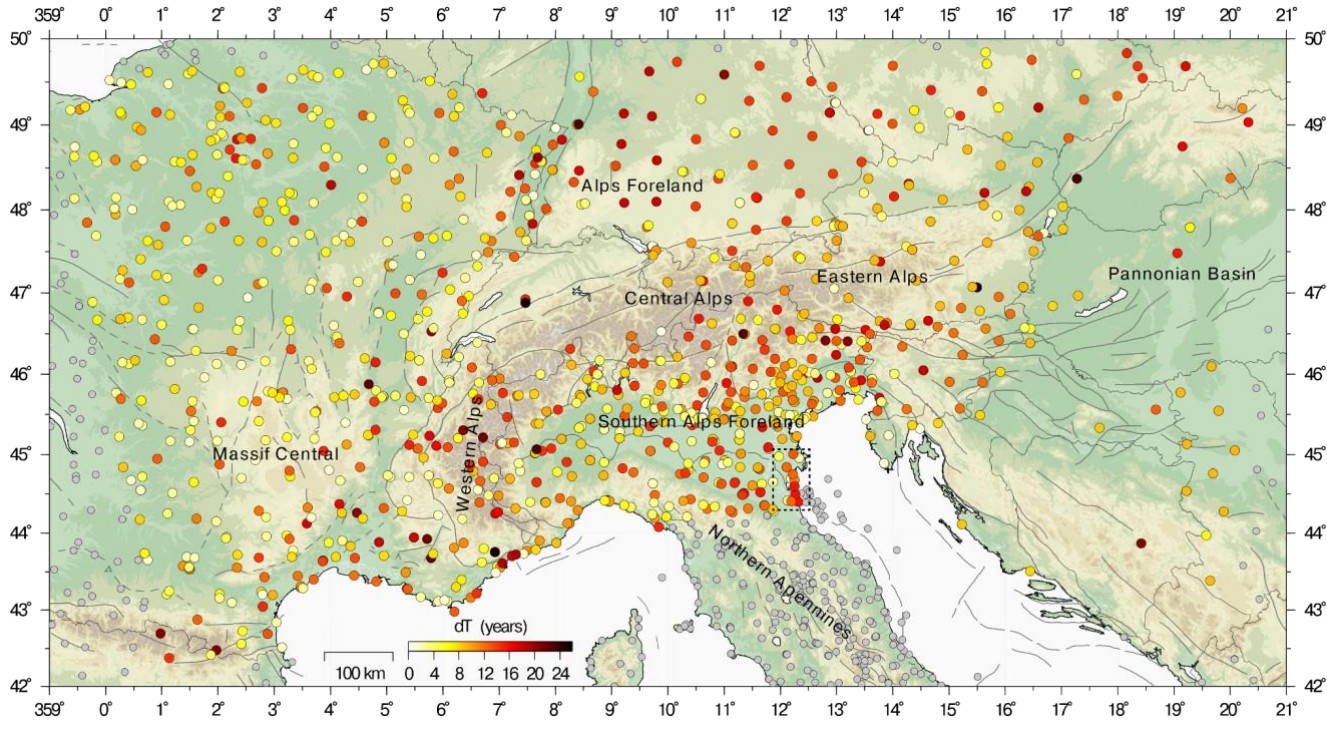

**Figure 1: Map of the study area showing the location of GNSS stations. Coloured circles show GNSS stations considered in the time-series analysis, with colours representing the length of the time-interval for which data are available at each station (0-25 years). The grey circles show GNSS stations not included in the time-series analysis to reduce contamination of deformation processes not associated with the Alps. Dark grey lines represent mapped faults from the Geodynamic Map of the Mediterranean. The dashed box includes GNSS stations affected by anthropogenic deformation signals (Palano et al., 2020).**

## 2 Methods for the spatially-correlated signals extraction in GNSS time series

Two widely used techniques for extracting CMS from a GNSS network are the Stacking Filtering Method (SFM, Wdowinski et al., 1997) and the Weighted Stacking Filtering Method (WSFM, Nikolaidis, 2002), which differs from the first because of a weighting factor based on the uncertainty associated with the GNSS data at each epoch.

Examples of time series filtering with the WSFM are provided by Ghasemi Khalkhali et al. (2021) in Northwest Iran, Jiang et al. (2018) in California and by Zhang et al. (2020) in China. The networks of the aforementioned studies span less than 1000 km. However, when considering networks covering larger areas, the assumption that the CMS has uniform spatial distribution throughout the network is not valid (Dong et al., 2006; Tian and Shen, 2016; Ming et al., 2017), and the stacking methods become imprecise.

To take into account spatial heterogeneities, Tian and Shen (2016) propose an alternative stacking approach: the Correlation-Weighted Spatial Filtering (CWSF) method. Unlike the SFM, CWSF includes the spatial variability of CMS through a

weighting factor, which depends on the correlation coefficient between the residual position time series and on the distance between the stations. Zhu et al. (2017) use CWSF to estimate the CMS on the Crustal Movement Observation Network of China and discuss the effects of the thermal expansion and environmental loading, which includes atmospheric pressure loading, non-tidal ocean loading and continental water storage. They find that while vertical CMS are mainly associated with environmental loading, thermal expansion plays a minor role.

A filtering method similar to CWSF, called CMC Imaging, is developed and used by Kreemer and Blewitt (2021) in western Europe to extract common mode components that are as local as possible. The main difference between CWSF and CMC Imaging is that the former uses as a weighting factor both the distance and the correlation coefficient among the stations, while the latter only the correlation coefficient, showing that it is representative of the distance among the stations. While the authors do not explore the nature of the extracted CMS, they show that the CMC Imaging method is very effective in filtering out CMS from GNSS time series, increasing the accuracy and precision of the velocity estimation. In particular, they show that the minimum length of a time series needed to retrieve the long term velocity, within a given confidence limit, is almost halved after the filtering.

Multivariate statistical techniques like Principal Component Analysis (PCA) and Independent Component Analysis (ICA) are filtering techniques based on a completely different approach than stacking. Since they allow to take into account for the spatial variability of CMS (Dong et al. 2006), ICA and PCA are used to characterize and interpret them. Multivariate statistics techniques are also applied to study spatially-correlated seasonal displacements, which have been the target of several researches in the last few years.

In California, Tiampo et al. (2004) associate a seasonal signal, extracted through the Karhunen-Loeve expansion technique, with the combined effect of groundwater and pressure loading. In Taiwan, Kumar et al. (2020) find a close relationship between atmospheric loading and CMS, extracted using a PCA; while Liu et al. (2017) apply a ICA to show that in the Nepal Himalaya region annual vertical displacements are associated with atmospheric and hydrological loading.

Yuan et al. (2018) use three Principal Components (PCs) for CMS filtering over China, because of the presence of spatial gradients related to the large extension of the study region. In that work, the authors show that environmental loading is one of the sources of the CMS and that vertical GNSS velocities uncertainties are significatively reduced (54%) after CMS filtering. Pan et al. (2019) find that the precision of the GNSS velocities, especially in the vertical component, increases after removing spatially-correlated signals related to draconitic errors and to climate oscillation (La Niña - El Niño). The spatially-correlated signals are identified by applying a PCA to the GNSS time series, where the linear trend and the seasonal signals are removed. Pan's work is a good example of how vertical displacements are more affected by climate-related processes and data processing errors than the horizontal ones, demonstrating that the vertical component is particularly worth analyzing with care.

The application of the ICA also proved effective for time series filtering, as shown by Hou et al. (2019): they identify spatially-correlated signals and even though they do not provide an interpretation, classifying them as CME, they show that the precision of the time series significantly increases after the filtering by ICA. Liu et al. (2015) use both PCA and FastICA algorithms

(Hyvärinen and Oja, 1997) to extract and interpret CMS as caused by atmospheric and soil moisture loading in the UK and the
Sichuan-Yunnan region in China.
Other examples of the influence of the non-tectonic processes on vertical velocity estimation are provided by Riddell et al.
(2020), who study the vertical velocities of the GNSS stations in Australia to estimate the contribution of the glacial isostatic
adjustment. One of the results of Riddel's work is the reduction of the vertical velocity uncertainty, achieved by first subtracting
the displacements associated with atmospheric, hydrological and non-tidal ocean loading from the GNSS time series, and then
filtering the residuals by applying both PCA and ICA.
The vbICA is a multivariate statistics-based blind source separation algorithm (Choudrey, 2002) implemented by Gualandi et
al. (2016) for solving the problem of blind source separation of deformation signals in GNSS position-times series and has
been successfully used to extract tectonic and hydrological transient deformation signals in (e.g., Gualandi et al., 2017a;
Gualandi et al., 2017b; Serpelloni et al., 2018). Larochelle et al. (2018) applied vbICA to study the relationship between GNSS
and Gravity Recovery and Climate Experiment (GRACE)-derived displacements in Nepal Himalaya and Arabian Peninsula,
with the goal of extracting seasonal signals and identifying the processes that generate them. Serpelloni et al. (2018) and Pintori
et al. (2021) use vbICA to characterize hydrological deformation signals associated with the hydrological cycle at a spatial
scale not resolvable by GRACE observations, separating ground water storage signals from other surface mass loading signals;
while Silverii et al. (2021) perform a vbICA decomposition on GNSS time series in the Long Valley Caldera region (California,
USA) to separate volcanic-related signals from other deformation processes, in particular the one associated with hydrology.
This method is also recently applied to InSAR data (Gualandi and Liu, 2021) to estimate the displacement caused by sediments'
compaction in San Joaquin Valley (California) and to separate a seasonal signal from the tectonic loading in the Central San
Andreas Fault zone.
**3 Data and Methods**
**3.1 GNSS dataset and time-series analysis**
Over the European plate, in particular, GNSS networks managed by national and regional agencies, provide a rather uniform
spatial coverage (e.g., https://epnd.sgo-penc.hu/ and https://gnss-epos.eu/). Figure 1 shows the distribution of continuous
GNSS stations operating across the great Alpine area where, excluding Switzerland for which raw observations are not
accessible, GNSS stations cover, rather uniformly, both the mountain range and the European and Adriatic forelands. We
analyze the raw GPS observations using the GAMIT/GLOBK (Ves. 10.71) software (Herring et al, 2018), following the
standard procedures of the repro2 IGS reprocessing scheme (http://acc.igs.org/reprocess2.html). This is part of a large
processing effort, including >4000 stations in the Euro-Mediterranean and African region, where sub-networks, made by <50
stations, dynamically and optimally selected based on daily data availability, are processed independently with GAMIT and
later tied together using common, sub-net, tie sites and IGb14 core-stations, using the GLOBK software. The details of the
processing are given in the Supplementary Information S1. The result of our analysis is a set of ground displacement time-

series, realized in the IGb14 reference frame (ftp://igs-rf.ign.fr/pub/IGb14). The resulting position time-series (hereinafter IGb14-time series) have been then analyzed in order to estimate, and correct, instrumental offsets due to changes in the station's equipment setup, as extracted from sitelog or RINEX file headers.

We consider the vertical displacement time-series of the stations between longitude 0°-21° and latitude 42°-50°N (see coloured circles in Fig. 1) in the 2010-2020 time-span, excluding the sites in the northern Adriatic coast, known to be affected by anthropogenic deformation signals (dashed box in Fig. 1) due to gas extraction (Palano et al., 2020) and the stations located in the northern and central Apennines, where other tectonic and geodynamic processes are going on. We focus on the last decade, in order to have the most uniform set of continuous measurements possible in, at least, a 10 years time-span. We acknowledge that some of the stations shown in Fig. 1 have much longer time-series, but this time-interval maximizes the number of simultaneous observations at many stations.

The IGb14 vertical displacement time-series are analyzed with the blind source separation algorithm based on vbICA (Choudrey and Roberts, 2003; Gualandi et al., 2016). This technique falls under the umbrella of the so-called unsupervised learning approaches, and it aims at finding statistically independent patterns that can be linearly combined to reconstruct the original dataset. Differently from other commonly used ICA approaches, like for example FastICA (Hyvarinen and Oja, 1999), the adopted vbICA is a modeling approach that uses a mix of Gaussians to reproduce the probability density functions (PDFs) of the underlying sources. The variational Bayesian approach introduces an approximating PDF for the posterior parameters of the model, and the cost function to be maximized is the Negative Free Energy of the model, which can be explicitly calculated once a specific form for the approximating posterior PDF is chosen. This framework is particularly advantageous because it allows for more flexibility in the description of the sources' PDF, giving the chance to model multimodal distributions and to take into account missing data in the input time series.

The input time-series contains a secular motion, roughly representing the vertical rate in the IGb14 reference frame, which is superimposed by a variety of signals, of different temporal and spatial signatures. The first step of our analysis is to estimate a linear component to represent the secular motion and remove it from the time series. This is required by the fact that the vbICA is more effective in separating the sources when the temporal correlation in the dataset is low. Here, rather than using a classic trajectory model (e.g., Bevis and Brown, 2014) to model and detrend the original time-series, in order to avoid biases in the estimates of station velocities due to the short length of the time series and to the possible presence of strong nonlinear signals, we take this step in a multivariate sense as in Pintori et al. 2021. We perform a first ICA decomposition considering 8 components (or ICs). The number of components is determined by applying an F-test to establish if a more complicated model is supported by the data at a 0.05 significance level (Kositsky and Avouac, 2010). The results of this analysis are reported in Fig. S1, and show that one component, nominally IC2, contains a linear trend, with some cross-talk with a seasonal (annual) signal, as shown in Fig. 2.

Before discussing the vbICA results, we briefly explain how to interpret the temporal evolution and the spatial distribution of the ICs, so that it is possible to retrieve the displacements associated with them. The color of each GNSS site in Fig. 2 represents the IC2 spatial response (U2), which indicates the maximum displacement associated with the IC2, while the temporal function

V2 is normalized between 0 and 1. The displacement associated with IC2 between two epochs (e.g. $t_1$ and $t_2$, with $t_2 > t_1$) at the
station n is computed as $V1(t_2)*U1_n - V1(t_1)*U1_n(t_1)$, where $V1(t_2)$ is the value associated with the temporal evolution of the IC
at the epoch $t_2$. $U1_n$ depends on the site, but not on the epoch; its unit of measurement is mm, while V has no units of
measurement. As a result, $V1*U1_n$ is in mm. It follows that if $U1_n$ is positive, as we observe for each station, and V1 is
increasing ($V1(t_2) > V1(t_1)$), the stations move upward during the $t_2 - t_1$ time interval. On the other hand, if $V1(t_2) < V1(t_1)$ the
stations move downward during $t_2 - t_1$. As regards Fig. 2, assuming $t_1 = 2010.0$ and $t_2 = 2020.0$, the displacements associated with
IC2 are ~30 mm upward at the "red" GNSS stations, ~30 mm downward at the "blue" GNSS stations and ~0 mm at the white
ones.

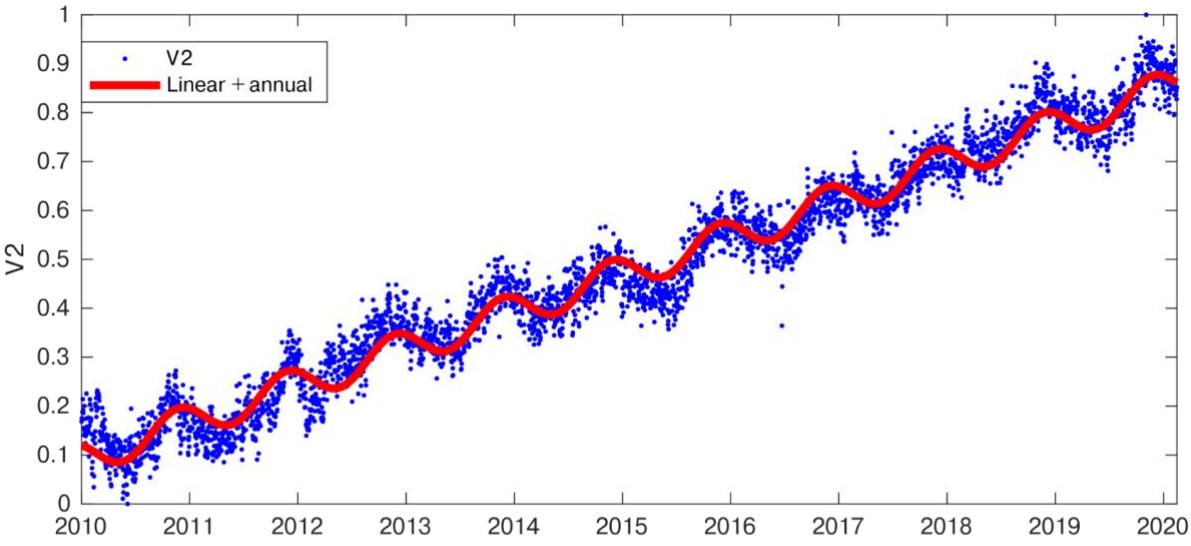


**Figure 2: Temporal evolution and spatial response of the IC2 of the GNSS decomposition. Time series have been corrected only for**

**instrumental offsets.**


We fit a linear trend to the temporal evolution of IC2 (V2) using the function

$V2(t) = q + m \cdot t + A \cdot sin\,(2\pi \cdot t + \varphi)$                                                                    (1)

Once estimated $m$ and $q$ from (1) via a non-linear least square approach, we compute the displacements associated with IC2,
considering as its temporal evolution the function y=$q + m \cdot t$; then, we remove the computed displacements from each
original, IGb14, time series, obtaining the detrended dataset used in the subsequent decomposition step. The advantage of this
approach, compared to a trajectory model, is that it is not necessary to assume any temporal evolution of the deformation
signals a priori, except for the limited number of functions that make up Eq. (1). This is particularly advantageous in cases
where either transients of unknown origin or amplitude and/or phase fluctuations of the seasonalities are affecting some stations
and could lead to a mismodeling by a trajectory model. Notice in particular how signals potentially biasing the linear trend,
like the multi-annual ones in case of short time series, are separated from the IC representing the stations' velocities.
The results of the vbICA applied to the detrended time-series are shown and discussed in Sect. 4.1.
**3.2 Meteo-climatic datasets**
The results of the decomposition of the geodetic dataset are compared with the results obtained from the analysis of
displacement time-series associated with different meteo-climate forcings. In particular, here we consider hydrological,
atmospheric loading and precipitation from global, gridded, models. These time-series are analyzed with the vbICA method
already used for the geodetic dataset, and the results are compared in Sect. 3.2.
The Land Surface Discharge Model (LSDM), developed by Dill (2008), simulates global water storage variations of surface
water in rivers, lakes, wetlands, and soil moisture, as well as from water stored as snow and ice. The LSDM is forced with
precipitation, evaporation, and temperature from an atmospheric model developed by the European Centre for Medium-Range
Weather Forecasts (ECMWF). Using the Green's function approach, Dill and Dobslaw (2013) compute daily surface
displacements at 0.5° global grids caused by LSDM-based continental hydrology (hereinafter HYDL), and by non-tidal
atmospheric surface pressure variations (hereinafter NTAL). We also considered the *École et observatoire des sciences de la*
*terre* (EOST) loading service, which provides a model for the atmospheric and hydrological loading induced displacements.
Ground displacements are computed using the Load Love Numbers estimate from a spherical Earth model (Gegout et al.,
2010). The atmospheric loading is modeled using the data of the ECMWF surface pressure, assuming an Inverted Barometer
ocean response; the hydrological loading includes soil moisture and snow height estimated from the Global Land Data
Assimilation System (GLDAS/Noah; Rodell et al., 2004). All the datasets we have considered are provided in the center of
figure reference frame, have daily temporal resolution and spatial resolution of 0.5°. It is worth noting that neither LSDM-
based nor EOST models consider deep groundwater variations. GRACE data are often used to study hydrologically-induced
deformation associated with groundwater; in fact, through the analysis of the gravity field variations, it is possible to retrieve
changes through time of the water masses. GRACE has the advantage of being influenced by groundwater variations, which
are not taken into account by the HYDL model, but at the cost of a lower temporal (i.e., monthly) and spatial (~300 km)
resolution.
The precipitation data we use are provided by the NASA Goddard Earth Sciences Data and Information Services Center
(Huffman et al., 2019), they are daily with a spatial resolution of 0.1°.

**4 Results**

**4.1 Decomposition of GNSS time-series**

Figure 3 shows the result of the vbICA decomposition on the detrended displacement time-series, using 7 components as
suggested by the F-test.
IC1 is a spatially uniform signal characterized by an annual temporal signature, as shown by the power spectral density (PSD)
plot in Fig. 3a.
The mean of the maximum amplitudes is 26 mm, while the histogram showing the distribution of displacement amplitudes is
shown in Fig. S4a.
IC2 shows a spatial response characterized by a clear E-W gradient, but, differently from IC1, its temporal evolution has not
a dominant frequency. The spatial response U2 of the eastern stations (in blue) is mainly negative, while the U2 of the western
stations (in red) is mainly positive. This means that when V2 is increasing the western (red) stations move up, while the eastern
(blue) ones move down. The sites in the central portion of the study area (in white) are very slightly affected by the IC2
component. The features of IC3 are analogous to those of the IC2, with the exception that a N-S gradient is present. The mean
of the amplitude of the absolute value of IC2 spatial distribution is 6.7 mm; and it is 5.6 mm for IC3. The histogram showing
the distribution of the absolute value is shown in Fig. S4b and S4c.
IC4 is an annual signal, as IC1, but with a heterogeneous spatial response: while some stations move upward some others
move downward. The mean of the amplitudes absolute value of the displacements is 2.7 mm; the relative histogram is shown
in Fig. S4d. The distribution of stations displaced with this phase difference seems to be mostly affected by geographical
features: the stations located in mountain regions subside when V3 increases, whereas the stations far from relief move upward.
The remaining three components are likely associated with local processes and discussed in the Supplementary Information
S3.

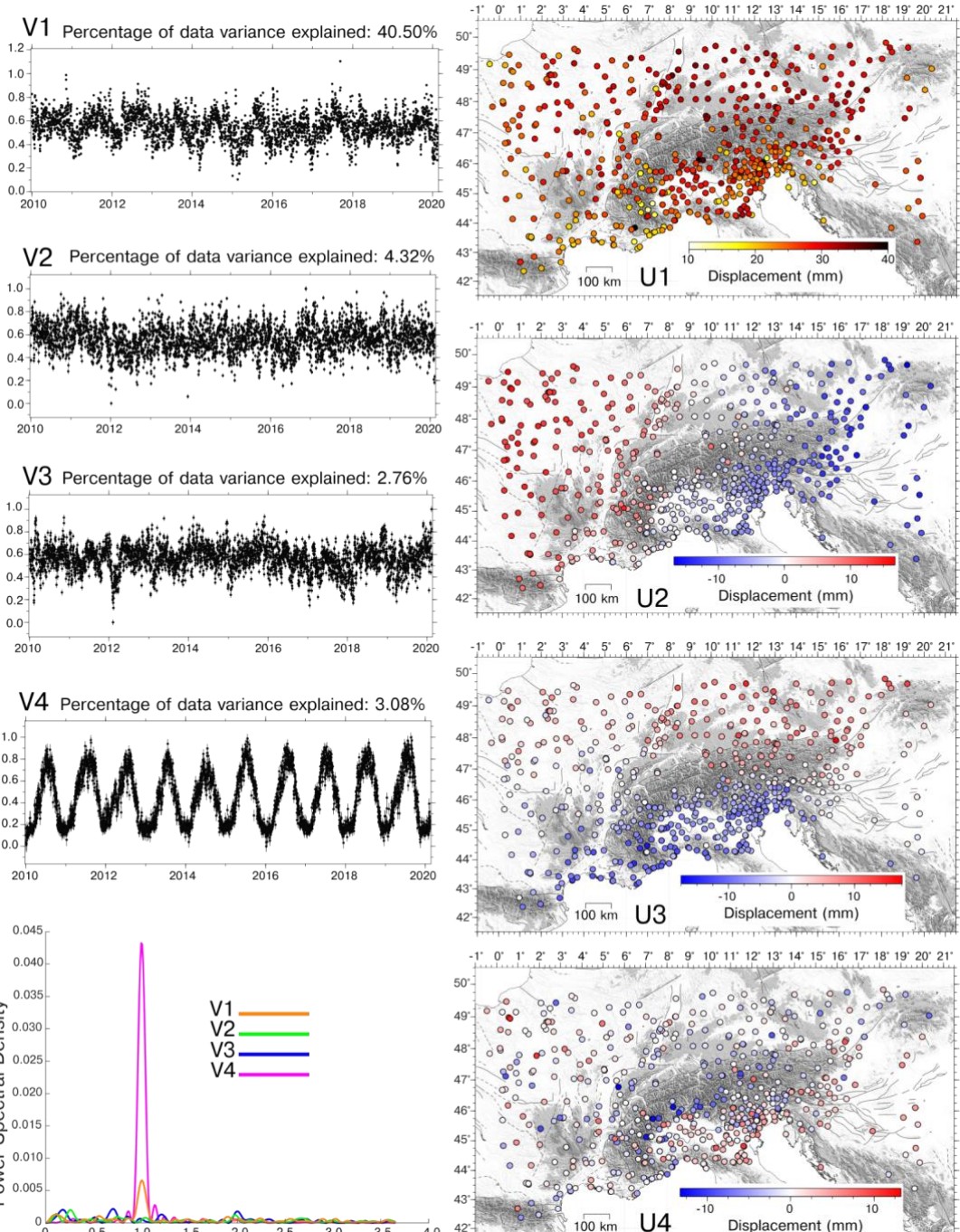

**Figure 3:** Temporal evolution, power spectral density and spatial response of: a) IC1; b) IC2; c) IC3; d) IC4.

## 4.2 GNSS vs environmental-related displacements

As discussed in the introduction, atmospheric and hydrological loading are likely the main sources of vertical displacement in the great Alpine region. Since they are both uniform in terms of spatial response, showing smooth spatial variations, we decided to check if the first 3 ICs of the GNSS decomposition are associated with the displacements due to atmospheric and hydrological loading, and with their pattern of variability.

The vbICA analysis separates the data into statistically independent signals, which is useful because independent signals are often caused by different and independent sources of deformation. Nonetheless, a single source of deformation, such as atmospheric or hydrological loading, can be spatially heterogeneous and characterized by peculiar spatio-temporal patterns. In this case, the vbICA separates a single source of deformation in different components associated with different spatio-temporal patterns. As a consequence, we decided to apply a vbICA decomposition on HYDL and NTAL model displacement time series in order to check if they show any pattern and if they resemble the spatial distribution of IC1, IC2 and IC3 of the GNSS decomposition. NTAL and HYDL data have not been detrended.

We analyze with vbICA the hydrological loading (HYDL) and atmospheric pressure (NTAL) induced ground displacement models (EOST and LSDM-based), in order to characterize the spatial pattern and temporal response associated with these deformation sources, and study any possible link with the geodetic deformation signals described in Sect. 4.1. We use the results of the global models to estimate the hydrological loading, even though we are aware that some local effects might not be captured. In fact, considering the extension of the study area, it is very complicated to take into account the local features needed to estimate the hydrological loading with a better precision than the one provided by the global models.

In particular, in this section we show the results obtained using the LSDM-based models because they take into account the water stored in rivers, lakes and wetlands, while the EOST models do not. The results obtained using the EOST models are presented in the Supplementary Information S2. Figure 4 and 5 show the spatial response, the temporal evolution and the PSD of the ICs obtained using three components, to the NTAL (4) and HYDL (5) ground displacements. We decided to use three components to reproduce the displacement patterns of IC1, IC2 and IC3 of the GNSS decomposition.

The first IC of both NTAL and HYDL shows a uniform spatial response, as IC1 of the GNSS dataset (Fig. 3a). The mean/median amplitude of the maximum displacements associated with NTAL is very similar to GNSS both in terms of mean/median amplitude (Table S1a) and distribution (Fig. 6, a); while for the HYDL model the amplitude is about two times smaller than NTAL.

IC2 and IC3 of both NTAL and HYDL show E-W and N-S gradients in the spatial response, respectively, as observed for IC2 and IC3 of the GNSS dataset (Fig. 3b, d). Since the ICs spatial response of the NTAL and HYDL decomposition are very similar, we also consider the sum of the displacement associated with NTAL and HYDL models, which can be considered as "environmental loading": we use the notation NTAL+HYDL_ICn to indicate the sum of the displacement associated with the n-th component of the NTAL and HYDL decomposition. The amplitude of NTAL+HYDL_IC1, NTAL+HYDL_IC2 and

NTAL+HYDL_IC3 are only slightly lower than the ones of GNSS_IC1, GNSS_IC2 and GNSS_IC3, as shown in Fig. 6
(panels g,h,i) and in Table S1a.
Concerning the temporal evolutions, IC1 of the HYDL model is an annual signal, while the IC2 and IC3 PSD plots indicate
the presence of multi-annual signals. Unlike the HYDL decomposition, all the ICs of the NTAL decomposition contain the
annual frequency, in particular IC2, whereas IC3 also contains semiannual ones. It is also worth noting that the temporal
evolution of the ICs associated with the NTAL model are much more scattered than the ones resulting from HYDL, clearly
indicating that the displacements due to atmospheric pressure variations can show large fluctuations at daily timescale.
We also perform a vbICA decomposition on both datasets using two and four components, presented in the Supplementary
Information (Fig. S6 and S7). When using only two ICs, the results obtained (Fig. S6) are very similar to the first two ICs of
the 3-components decomposition. The first three ICs of the four component decompositions (Fig. S7) have both temporal
evolution and spatial distribution very similar to what is shown in Fig. 4 and Fig. 5. IC4 of the NTAL model has an annual
signature and a E-W gradient with a shorter wavelength compared to IC2, while IC4 of the HYDL decomposition has a NW-
SE gradient. This suggests that the N-S and E-W spatial patterns associated with the meteoclimatic datasets are a robust feature,
being insensitive to the number of components chosen in the decomposition. It is also worth noting that the decompositions of
the NTAL and HYDL models explain the 98.89% and the 97.03% of the total variance when using 3 ICs, suggesting that
increasing the number of the ICs is not necessary. As a result, in the following discussion we refer to the results obtained from
the 3-components decomposition using the LSDM-based models, but remember that the results obtained using the EOST
models are fully comparable (Supplementary Information S2).

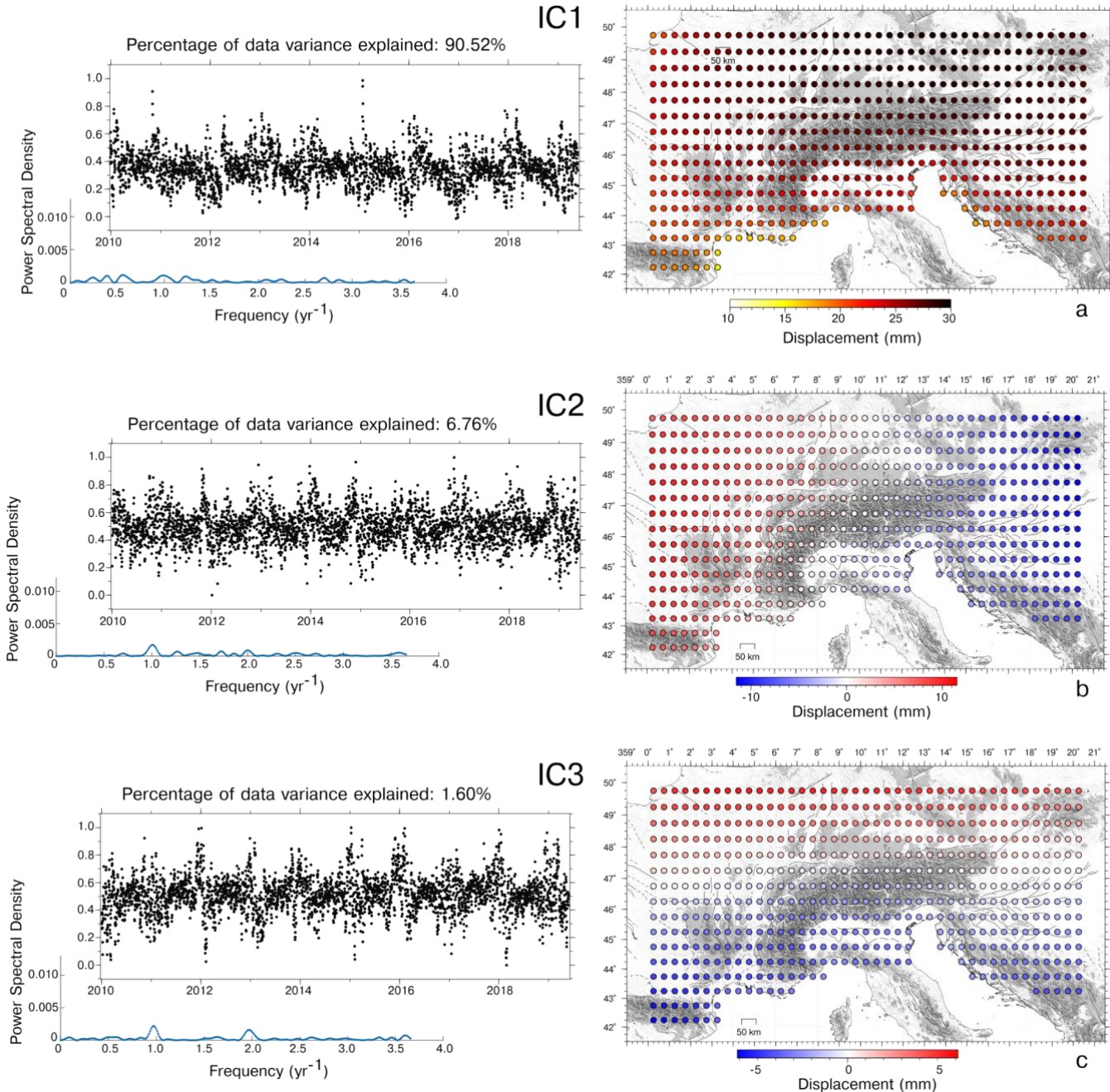


Figure 4: Temporal evolution, power spectral density and spatial response of IC1, IC2, IC3 of the NTAL model.

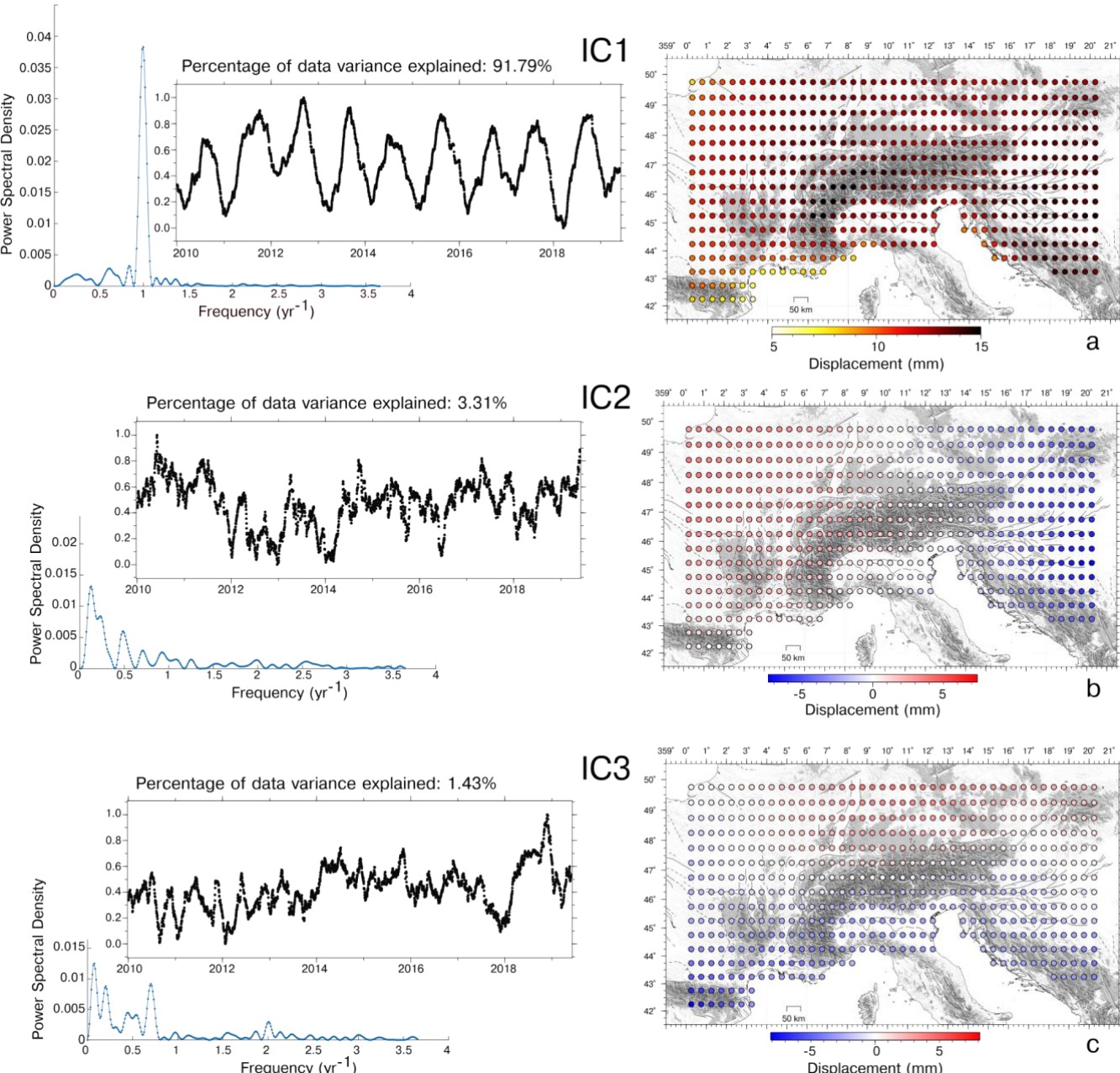


**Figure 5: Temporal evolution, power spectral density and spatial response of IC1, IC2, IC3 of the HYDL model.**


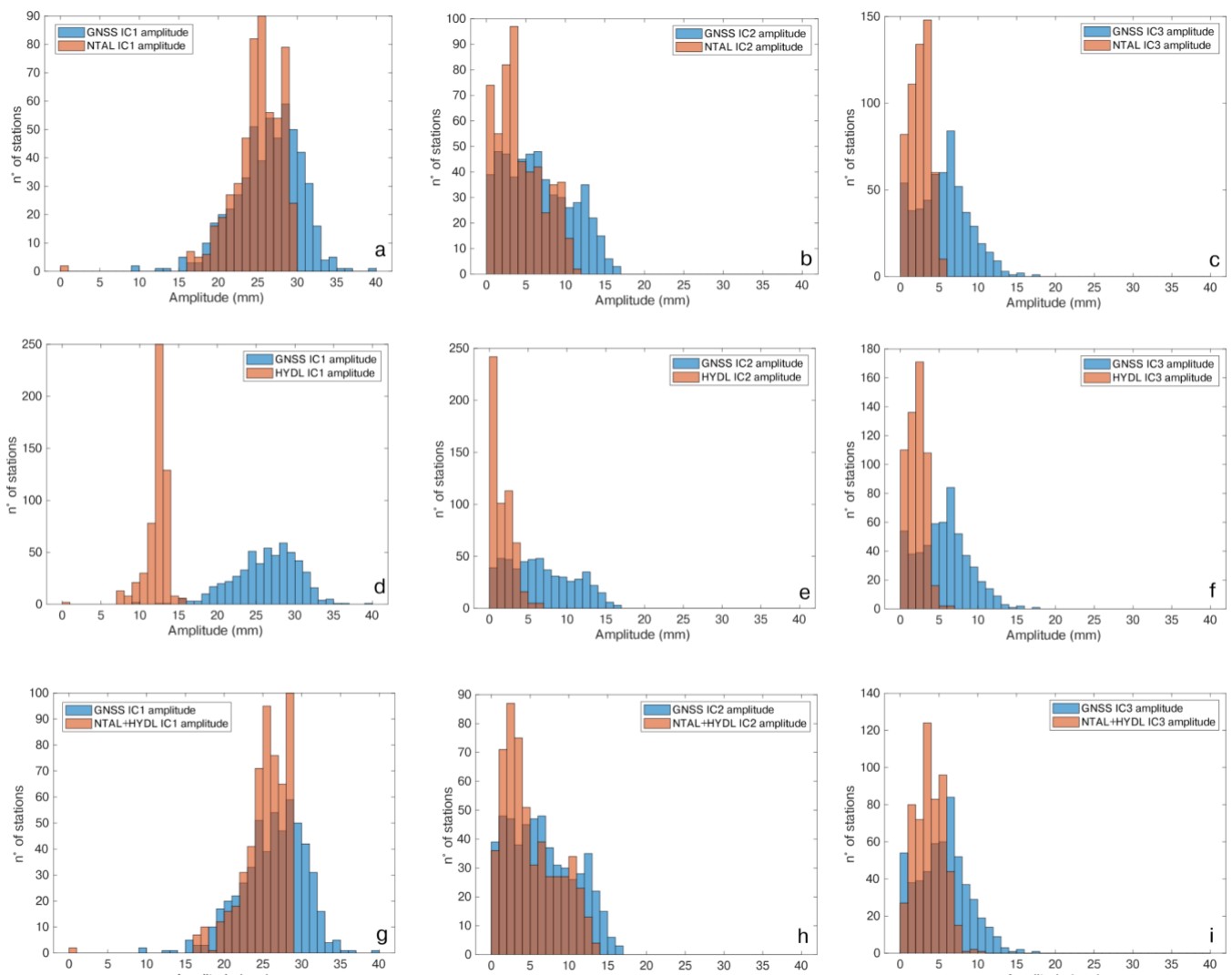

**Figure 6. Histogram of the maximum displacements associated with:**

**(a) IC1 of the NTAL decomposition (orange), compared with the IC1 of the GNSS decomposition (blue); (b) same as (a) but considering IC2; (c) same as (a) but considering IC3;**

**(d) IC1 of the HYDL decomposition (orange), compared with the IC1 of the GNSS decomposition (blue); (e) same as (d) but considering IC2; (f) same as (d) but considering IC3;**

**(g) IC1 of the NTAL+HYDL decomposition (orange), compared with the IC1 of the GNSS decomposition (blue); (h) same as (g) but considering IC2; (i) same as (g) but considering IC3.**

In order to quantify the agreement between the displacements associated with the hydrological and atmospheric pressure loading and the ICs of the GNSS dataset displaying consistent spatial patterns (IC1, IC2, IC3), we compute, for each GNSS station, the Lin concordance correlation coefficient (Lin, 1989) between the displacement reconstructed by the ICs associated

with the different LSDM-based models. Unlike Pearson's correlation coefficient, Lin's one takes into account similarities on both amplitudes and shapes of two time series.

The IC1 of the GNSS decomposition (GNSS_IC1) is compared with the first component of both NTAL (NTAL_IC1) and HYDL (HYDL_IC1) datasets by associating each GNSS site with the nearest grid-point where NTAL and HYDL displacements are computed.

When considering the NTAL_IC1, we observe (Fig. S8a) a high temporal correlation with GNSS_IC1, while the correlation between GNSS_IC1 and HYDL_IC1 is significantly lower (Fig. S9a). In both cases the value of the Lin correlation coefficient is quite uniform in the dataset (~0.59 for NTAL_IC1 and ~0.35 for HYDL_IC1). The Pearson correlation is similar to Lin's one (0.60 for NTAL_IC1 and 0.35 for HYDL_IC1), indicating that the amplitude of both NTAL_IC1 and HYDL_IC1 is similar to the GNSS_IC1 amplitude. It is worth noting that if we consider NTAL+HYDL_IC1, the correlation with GNSS_IC1 increases to ~0.73 (Fig. 7a). As a result, we can interpret GNSS_IC1 as the combined contribution of NTAL and HYDL, where NTAL plays the dominant role.

When considering IC2, we observe similar correlations between GNSS_IC2 and either NTAL_IC2 or HYDL_IC2 (Fig. S8b, S8b). Nonetheless, in this case the correlation patterns are less uniform than the IC1 case, and few stations are even negatively correlated with both NTAL_IC2 and HYDL_IC2 displacements. The sites where GNSS_IC2 displacements are negatively or weakly correlated with NTAL_IC2 are the ones with the lowest IC2 amplitude. In fact, if we consider the stations whose maximum displacements associated with GNSS_IC2 are larger than 3 mm, which are 411 out of 545, their mean Lin correlation with NTAL_IC2 is 0.52; while the stations with amplitudes smaller than 3 mm have a mean correlation of 0.17. This is due to the fact that, given the low displacements associated at these stations, the correlation is more sensitive to noise. The agreement between the GNSS_IC2 and NTAL_IC2 is also confirmed by the Pearson correlation coefficient between the temporal evolution of the two ICs, which is 0.63; while the Pearson correlation between GNSS_IC2 and HYDL_IC2 is 0.28. The same pattern is observed when comparing GNSS_IC2 with NTAL+HYDL_IC2 (Fig. 7b): using 3 mm as threshold between large and small GNSS_IC2 maximum displacements, the mean correlation is 0.57 for the stations most affected by this signal and 0.14 for the remaining ones. This suggests that also GNSS_IC2 is likely related to NTAL and HYDL loading processes.

The Lin correlation between GNSS_IC3 and NTAL+HYDL_IC3 resembles what just shown for IC2 (Fig. 7c): at sites where the GNSS_IC3 maximum amplitude is larger than 3 mm, which are 414 out of 545, the mean correlation with NTAL+HYDL_IC3 is 0.44; while it is 0.10 for the remaining ones. As for IC1, both GNSS_IC2 and IC3 displacements are best reproduced when considering the combined effect of NTAL and HYDL (see Fig. S8c, S9c compared to Fig. 7). The Pearson correlation between GNSS_IC3 and NTAL_IC3 is 0.47; while between GNSS_IC3 and HYDL_IC3 is 0.30.

To summarize, the three common mode signals components of the GNSS decomposition (IC1, IC2, IC3) are likely due to the combined effect of the atmospheric and hydrological loading. Due to the similarity between the spatial response of displacements associated with these two processes, it is possible that the vbICA technique is not able to separate them in the geodetic data; nonetheless, it highlights their spatial variability through IC2 and IC3.

Examples of comparison between climate-related displacements reconstructed at two different sites and the GNSS
decomposition are shown in Fig. 8.

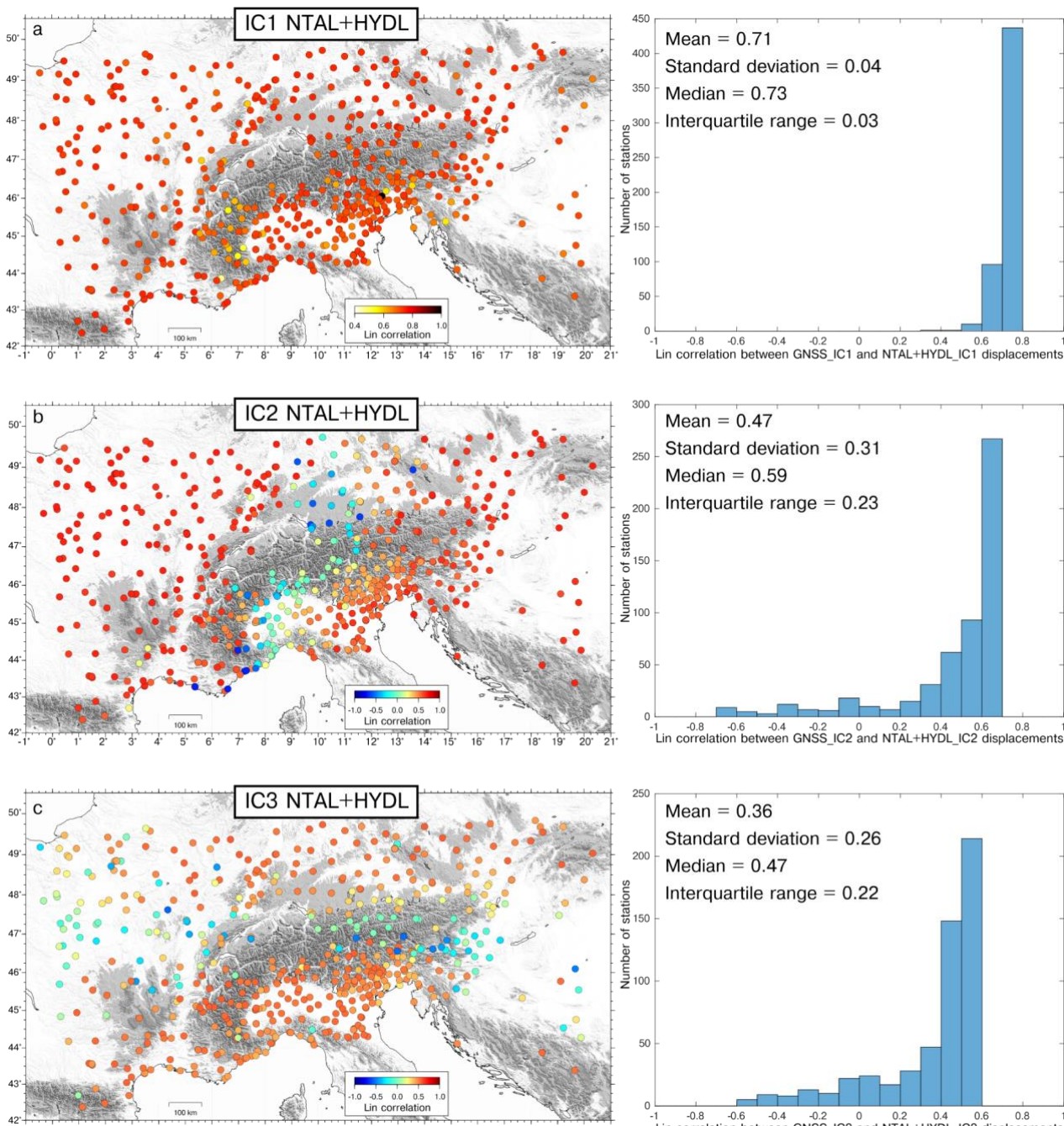

**Figure 7: Lin correlation coefficients between: a) GNSS-IC1 and NTAL+HYDL_IC1; b) GNSS_IC2 and NTAL+HYDL_IC2; c)**
**GNSS-IC3 and NTAL+HYDL_IC3. Histograms of the correlation coefficients are also reported.**

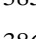

**Figure 8: Comparison, at the LYSH (Lon: 18.45°; Lat: 49.55°) site, between the displacements associated with: a) GNSS_IC1 and NTAL+HYDL_IC1; b) GNSS_IC2 and NTAL+HYDL_IC2 ; c) GNSS_IC3 and NTAL+HYDL_IC3. d), e), f) are the same as a), b), c), respectively, for the STV2 (Lon: 6.11°; Lat: 44.57°) site. A 30-days moving average filter is applied to better visualize the data.**

Concerning IC4 of the GNSS decomposition, it describes vertical motions in phase, and very well correlated, with the daily mean temperature of the investigated area (Fig. 9). Temperature data are provided by the E-OBS dataset from the EU-FP6 project UERRA (https://www.uerra.eu; Cornes et al., 2018). From the point of view of the spatial distribution of this component, most of the stations located in the mountain chain subside when the temperature increases, while the remaining stations uplift as the temperature increases. Figure S15 shows some cross sections plotting the maximum vertical displacements associated with IC4 together with topography, showing this peculiar spatial pattern.

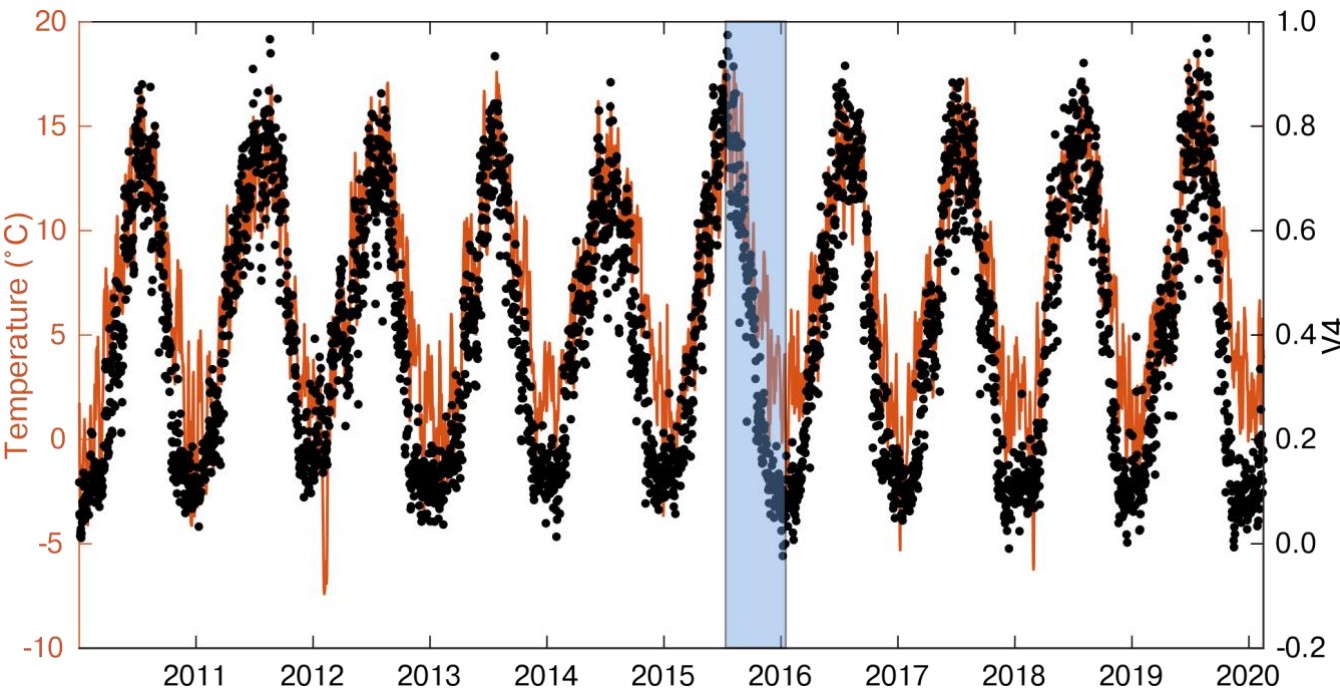

**Figure 9: Comparison between the daily mean temperature of the study area (orange) and the temporal evolution of IC4 (black dots). The shaded area represents the time interval associated with the maximum displacements shown in Fig. S15.**

**4.3 Vertical ground motion rates and noise analysis**

We show the impact of the filtering on GNSS displacement rates and uncertainties, where the filtered time-series are the result of subtracting from the IGb14-time series the combined displacement associated with the first 4 ICs discussed in Sect. 4.1, which represent the combined effect of the seasonal processes in phase with temperature and of the atmospheric and hydrological loading. We refer to these corrected time series as ICs filtered time series.

Velocities and uncertainties are estimated using the Hector software (Bos et al., 2013), assuming a priori noise models. Noise is commonly described as a power-law process

$$P_x(f) = P_0(f/f_0)^k \tag{2}$$

where $P_x$ is the power spectrum; $f$ the temporal frequency; $P_0$ and $f_0$ are constants; $k$ is the spectral index and it indicates the noise type.

If the power spectrum is flat (i.e., all frequencies have the same power), then the errors are statistically uncorrelated from one another, the spectral index is zero and the noise is called "white". Otherwise the noise shows a dependency with the frequency content, and it is referred to as "colored". In GNSS time series it has been typically observed the presence of noise with a power spectrum reduced at high frequencies, with the most popular models being a mix of random walk or "red" noise ($k = -2$) and flicker or "pink" noise ($k = -1$). Red noise is typically associated with station-dependent effects, while pink noise can be associated with mismodeling in GNSS satellites orbits, Earth Orientation Parameters (Klos et al., 2018) and spatially-

correlated large-scale processes of atmospheric or hydrospheric origin (Bogusz and Klos, 2016). Flicker plus white noise
model is commonly used in the analysis of GNSS time-series (e.g., Ghasemi Khalkhali et al., 2021 and references therein).
In order to select the best noise model for the input time series, we test different combinations of noise models, choosing the
one with the lowest value of the Akaike Information Criterion (AIC) and of the Bayesian Information Criterion (BIC). In
particular we consider:
-    Flicker + white noise;
-    A general power-law ($k$ not assigned) + white noise (PL+WN);
-    Flicker + Random walk + white noise.

Following the AIC and BIC criteria, the preferred noise model is PL+WN, where the parameters of the noise model (i.e., the
spectral index $k$) are estimated by the software using the Maximum Likelihood Estimation (MLE) method. MLE is also used
to estimate the station's rates and the associated uncertainties.
We then compare the vertical velocities, and their uncertainties, obtained before and after ICs filtering (Fig. 10). Although
annual and semi-annual signals are often included in the time series modeling, the displacements associated with the first four
ICs already contain these seasonal terms (Fig. 3). Consequently, the ICs filtered time series are modeled only with the linear
trend plus temporal correlated noise, while in the unfiltered time series modeling annual and semi-annual terms are also
included.
Fig. 11a shows histograms representing the differences in the vertical velocity estimates obtained from filtered and unfiltered
time-series. The differences are spatially quite homogeneous and of the order of tenths of mm yr$^{-1}$, with a median value of -
0.15 mm yr$^{-1}$. The velocity differences are almost entirely caused by the displacements associated with IC1, which have a
median rate of -0.12 mm yr$^{-1}$.
Concerning the uncertainties associated with the vertical velocity, the impact from ICs filtering is much more important (Fig.
10, f and Fig. S17): the initial median error is 0.30 mm yr$^{-1}$, the final 0.17 mm yr$^{-1}$.

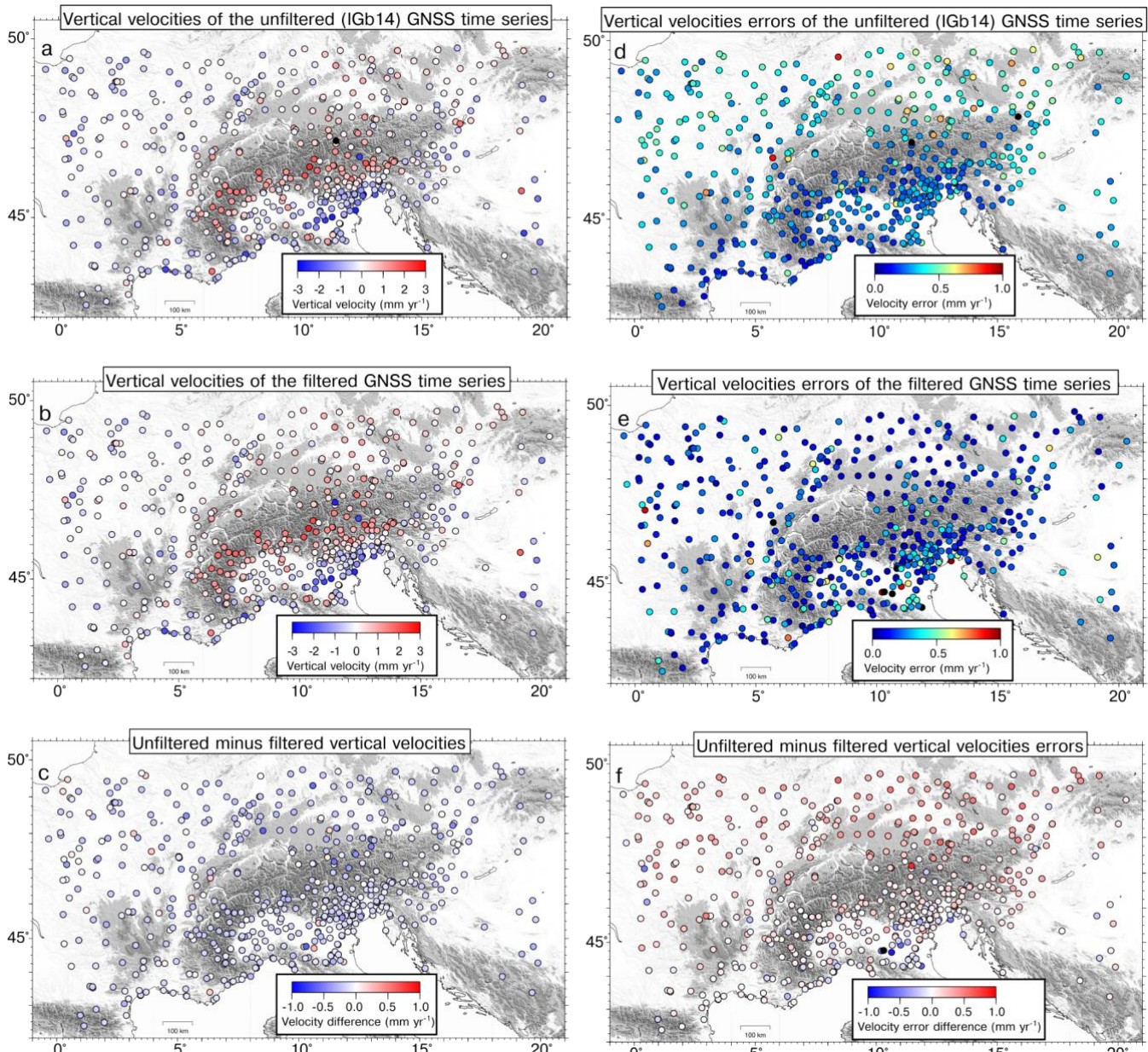

**Figure 10: a) Vertical velocities from the unfiltered GNSS time-series; b) vertical velocities from ICs filtered time series, obtained after subtracting the displacements associated with the first four ICs; c) difference between the velocities of panel a) minus velocities of panel b). d), e), f), same as a), b), c), but showing the error associated with the vertical velocities.**

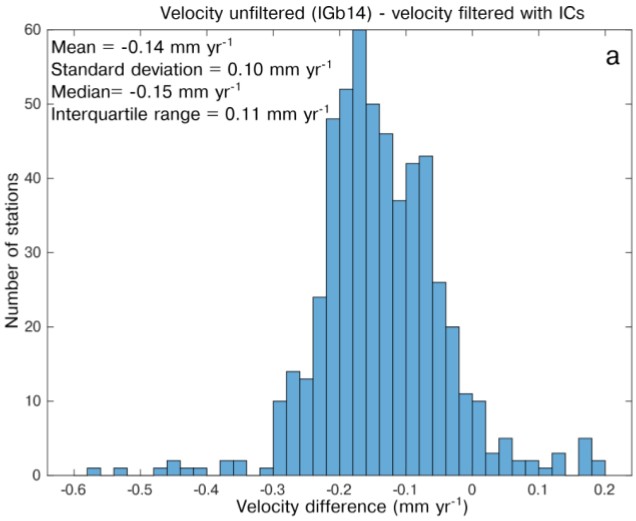

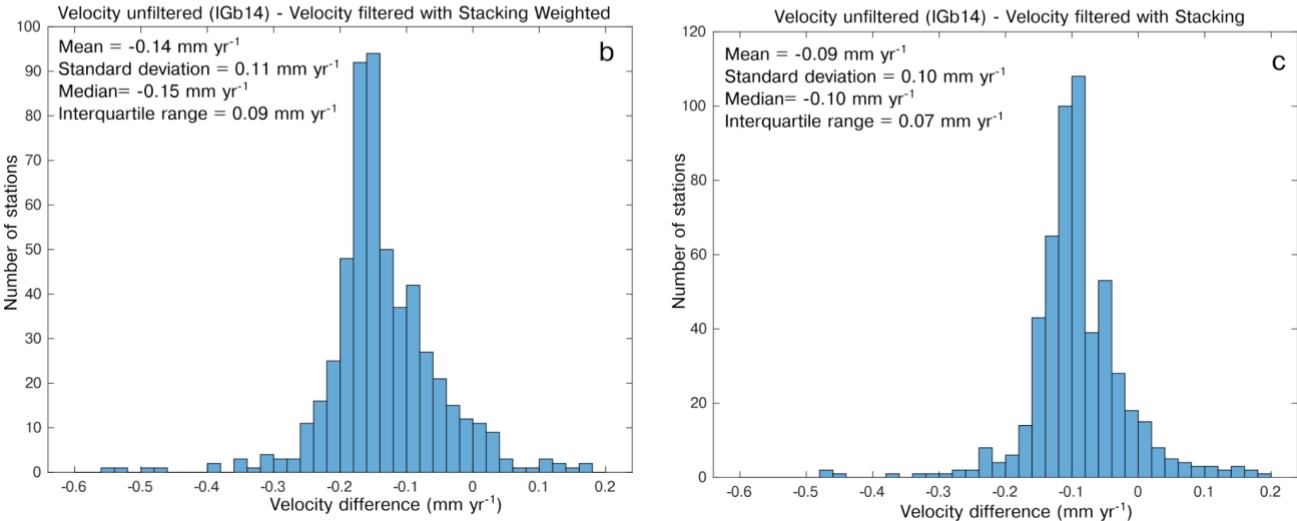

453

**Figure 11: Histogram of the difference between the velocity of the unfiltered time-series and the filtered ones using: a) the displacements associated with the first 4 ICs; b) the Weighted Stacking Filtering Method; c) the Stacking Filtering Method.**

The ICs filtering also has a strong impact on the noise characteristics. In fact, while in the unfiltered time series the percentage of white noise of the PL+WN model is negligible in most of the stations, it becomes dominant in the filtered ones (Fig. 12). This indicates that a large portion of the power-law noise is associated with the displacements described by the first 4 ICs, i.e. the atmospheric and hydrological loading and temperature-related processes.

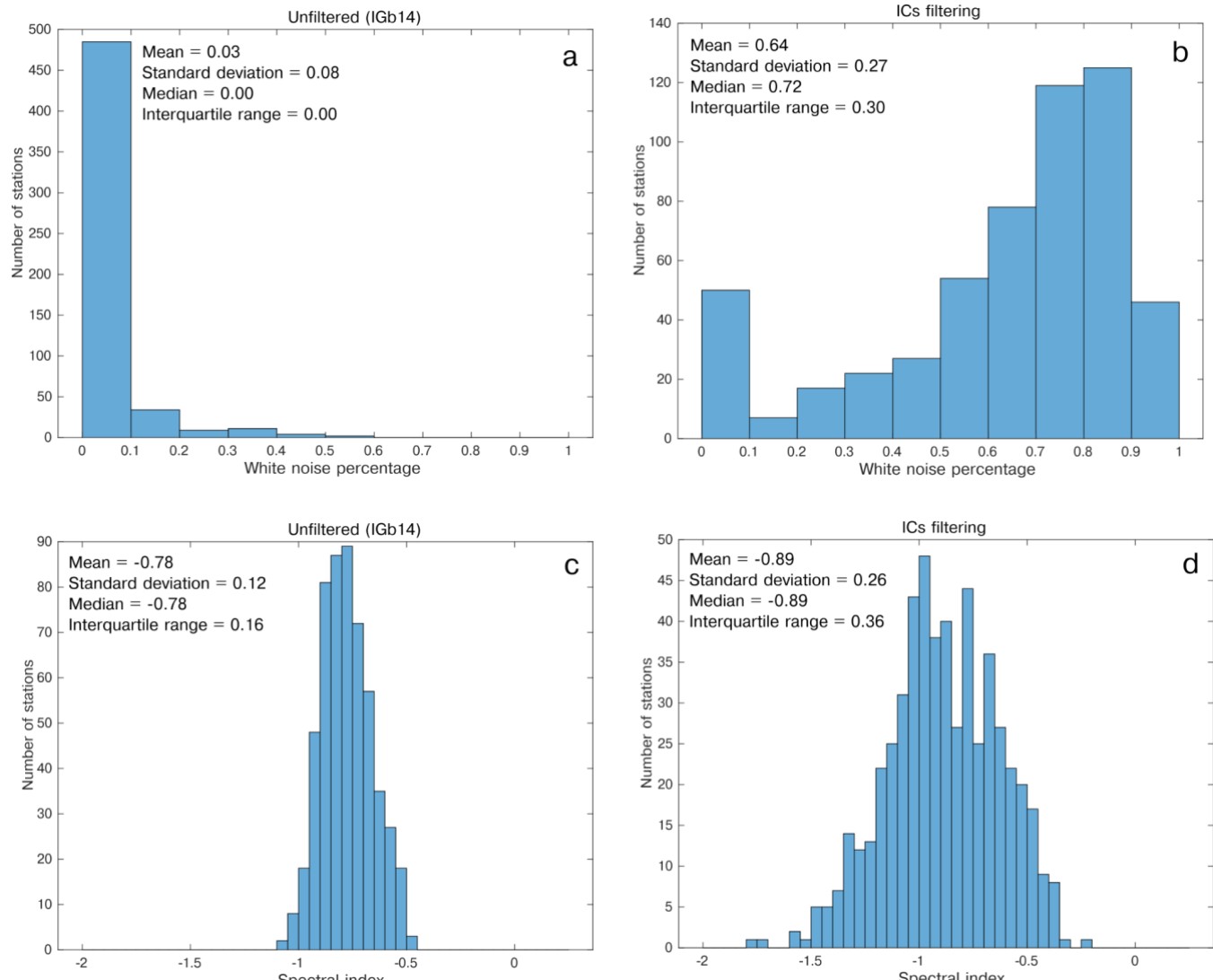

**Figure 12: Histograms of: (a) white noise percentage in the unfiltered time-series and (b) filtered time-series. (c), (d) same as (a) and (b) for the spectral index. The filtering is done by subtracting the displacements associated with the first 4 ICs.**

## 5 Discussion

### 5.1 Displacement time series filtering

Our goal is to estimate the vertical velocity of the GNSS stations associated with long-term geodynamic and tectonic processes, then we seek to remove signals associated with meteo-climatic processes. Instead of subtracting from the IGb14-time series the modeled displacements, such as those made available through loading services like GFZ, we prefer to subtract the displacements associated with the ICs. This approach minimizes biases due to the mismatch between the actual signal caused

by atmospheric and hydrological loading and the modeled ones. Larochelle et al. (2018) reached similar conclusions by comparing GRACE measurements and the results from ICA decompositions of GNSS displacements, which resulted to be more accurate in correcting GNSS from seasonal displacements than removing GRACE displacements, which smooth local effects in the data acquisition and processing. In order to support the approach followed, we estimated the scatter of the GNSS displacement time series by computing the mean standard deviation of 1) the time series given as input to vbICA (IGb14-time series), 2) the IGb14-time series minus the combined displacement associated with the first 3 ICs and 3) the IGb14-time series minus the displacements due to HYDL+NTAL from GFZ models. The resulting standard deviation is 5.32, 4.10 and 4.73, respectively. This demonstrates that removing the displacement associated with the first three ICs is more effective in reducing the scatter than removing the HYDL+NTAL contribution. Furthermore, in Fig. S19 we show that the filtering with HYDL+NTAL results in a smaller increase of the white noise percentage in the time series compared to the ICs filtering.

Considering that the stacking methods are widely used to estimate and remove CMS and CME from GNSS time-series (see Sect. 2), we compare the results obtained adopting the SFM and WSFM methods with the output of vbICA, in particular with the displacements associated with IC1 (Fig. 3a), which is clearly a CMS, given its homogeneity in its spatial response. CMS with the stacking methods is estimated using the GNSS_TS_NRS code (He et al., 2020) and it is compared with the displacements associated with IC1 estimating the Lin correlation coefficient. Figure 13 shows that there is an almost-perfect agreement between the IC1-related displacements and the CMS extracted with both stacking methods, suggesting that even simple approaches, such as SFM and WSFM, perform well at the scale of the study area.

We also estimate the vertical velocities of the GNSS stations after filtering the CMS using the two stacking methods. The rate differences between unfiltered and filtered time series have a median value of -0.15 and -0.10 mm yr$^{-1}$, using the WSFM and SFM, respectively (Fig. 11b, c). These values are close to the rates associated with IC1 displacements (median = -0.12 mm yr$^{-1}$), which are the primary cause of the velocity difference obtained from IGb14 and ICs filtered time-series, suggesting that the rate difference does not strongly depend on the filtering method adopted. As already shown in Sect. 4.3, the errors associated with the velocities of the unfiltered and filtered time series, which have median values of 0.30 and 0.17 mm yr$^{-1}$, respectively, have about the same value of the velocity difference between filtered and unfiltered time series. It follows that the velocity differences are, from a statistical point of view, barely significant. Nonetheless, it is worth considering that, according to the LSDM-based model, the displacements resulting from the combined effect of hydrological and atmospheric loading have a negative rate (median = -0.11 mm yr$^{-1}$; Fig. S16c) in agreement with the rate observed for IC1 (V1 in Fig. 3), suggesting that environmental loading may cause a small subsidence, at least in the observed time-span, which is captured by IC1. However, the rates of the displacements due to hydrological loading are model-dependent: according to LSDM, they show a negative linear trend (Fig. S16b), as opposed to what is observed using the EOST model (Fig. S16e). As a result, the rates of the displacements due to atmospheric + hydrological loading computed using the EOST model are not in agreement with the rates of the IC1 displacements. This is most likely a consequence of the differences in modeling the hydrological loading-induced displacements; in particular, the EOST model takes into account only water stored as snow and soil moisture, whereas the LSDM model also includes the contribution of rivers, lakes and wetlands.

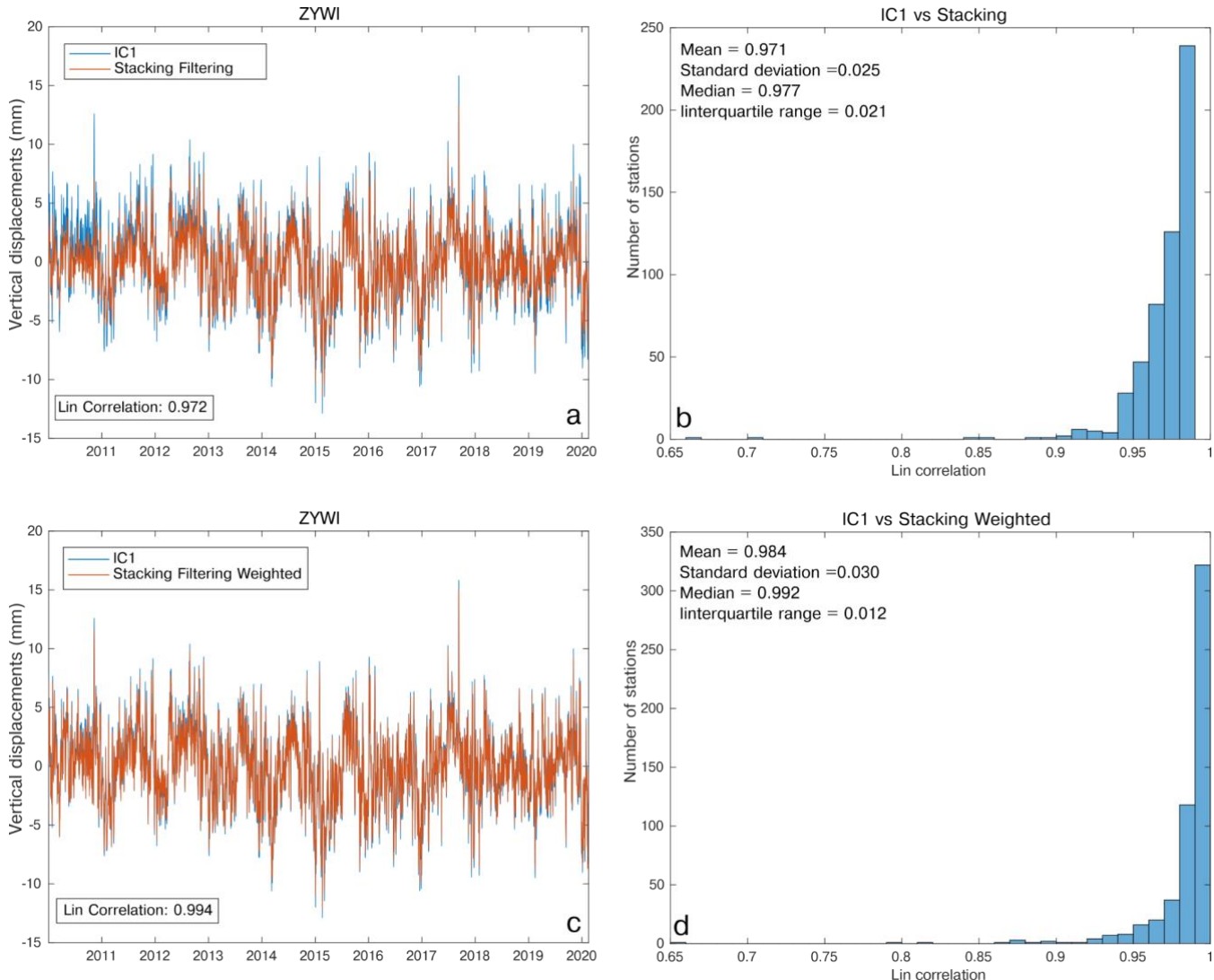

504

**Figure 13: Comparison between the displacement associated with IC1 at the ZYWI site and the CME estimated with**
**the Stacking Filtering Method (a) and the Weighted Stacking Filtering Method (c). We also show the histogram**
**representing the Lin correlation between the displacements associated with the IC1 and the CME estimated with the**
**Stacking Filtering Method (b) and the Weighted Stacking Filtering Method (d) at each site. We point out that the CME**
**computed with the aforementioned methods is, by definition, the same at each station; whereas the displacements**
**associated with IC1 have the same temporal evolution but (slightly) different amplitudes. We plot the station ZYWI as**
**an example.**

513 The stacking methods used to estimate the CMS are easier and faster to implement than the vbICA analysis. Depending on the

514 research target, these common mode signals might be worth removing, in order to obtain a more precise, and eventually

accurate, estimation of the GNSS linear velocities or retained to study, for example, seasonal deformation. Multivariate statistics and/or source separation algorithms applied to ground displacement time-series allow one to extract and interpret them in terms of the physics behind them, through a comparison with other displacement datasets or models. Furthermore, time series can be filtered not only from CMS, but also from signals associated with spatially uncorrelated processes, as we did in Sect. 4.3 estimating the vertical velocities filtered from non-tectonic processes related to the first four ICs.

In Sect. 4.3 we also show that the colored noise in the time series is significantly reduced by the ICs filtering. This result is in agreement with the results of recent studies conducted in other regions, such as Antarctica (Li et al., 2019) and China (Yuan et al., 2018). Both studies show that ICA or PCA filtering of GNSS time series suppress the colored noise amplitudes but have little influence on the amplitude of the white noise. Furthermore, Klos et al. (2021) analyzes the effect of atmospheric loading on the noise of GNSS stations in the European plate, finding that the noise is whitened when NTAL contribution is removed. The description of atmospheric processes at the scale of the Alps can be seen as small scale when compared, for example, to the circulation in the northern hemisphere. Small scale processes are usually interpreted as noise, but they may affect the large-scale dynamics (e.g., Faranda et al., 2017). It follows that these small scale processes should be represented with an appropriate stochastic formulation. Since the CMS are typically characterized by PL+WN noise, the link that we find between CMS and atmospheric and hydrological signals could provide a hint on the type of noise that is more suitable to describe such small scale perturbations when modeling the large-scale dynamics of the atmosphere.

**5.2 ICs interpretation**

Our analysis supports the interpretation that the displacements associated with IC1, IC2 and IC3 are likely due to the combined effect of the hydrological and atmospheric loading, whose spatial responses are not homogeneous over the study area. In support of this interpretation we can refer to Brunetti et al. (2006), who applied a PCA to precipitation data in the great Alpine area. They highlighted the presence of N-S and E-W gradients in the spatial response of meteo-climating forcing processes. The authors suggest that the main cause of the spatial and temporal variability of the precipitation is the North Atlantic Oscillation (NAO), which also causes fluctuation of the atmospheric pressure (Vicente-Serrano and López-Moreno, 2008). It is then likely that weather regimes like the NAO and the Atlantic Ridge, influence both NTAL and HYDL, which is mainly forced by precipitation, so that the spatial patterns of the ICs associated with atmospheric and hydrological loading are the same of NAO (N-S) and Atlantic Ridge (E-W). The vbICA algorithm is not able to separate NTAL and HYDL because they are not independent from a mathematical point of view. This emerges also from the recent work by Tan et al. (2022), who performed an ICA on GNSS time series of the Yunnan Province of China and interpreted IC1 as the average effects of the joint patterns from soil moisture and atmospheric-induced annual surface deformations. Let us consider for example the case of IC2_NTAL and IC2_HYDL. They have two different temporal evolutions (V2_NTAL and V2_HYDL); but the spatial distributions (U2_NTAL and U2_HYDL) have the same pattern, i.e. they only differ for a weighting factor $k$. Then, we can write U2_NTAL=$k$*U2_HYDL.

The displacement d resulting from the combined effect of IC2_NTAL and IC2_HYDL is then:

d= IC2_NTAL + IC2_HYDL= U2_NTAL*V2_NTAL + U2_HYDL*V2_HYDL= U2_HYDL*($k$*V2_NTAL+V2_HYDL).
As a result, the displacement due to IC2_NTAL + IC2_HYDL is identified by a single spatial distribution U2_HYDL and a
temporal evolution k*V2_NTAL+V2_HYDL. Then, if we do not make any prior assumptions about V2_NTAL and
V2_HYDL, it is not possible to separate IC2_NTAL and IC2_HYDL from a statistical point of view.
In Sect. 4.2 we show that not only IC2_NTAL and IC2_HYDL have very similar spatial patterns, but also IC1_NTAL and
IC1_HYDL,  IC3_NTAL and IC3_HYDL have similar spatial responses. Then, the GNSS time-series decomposition in the
Alpine area does not allow separating the effect of the hydrological loading from the atmospheric loading with an ICA
approach.
We also performed a vbICA analysis on precipitation data (RAIN) recorded over the study region, using 3 ICs (Fig. 14). The
spatial pattern of the ICs is analogous to the ones associated with NTAL and HYDL (Fig. 4 and Fig. 5).

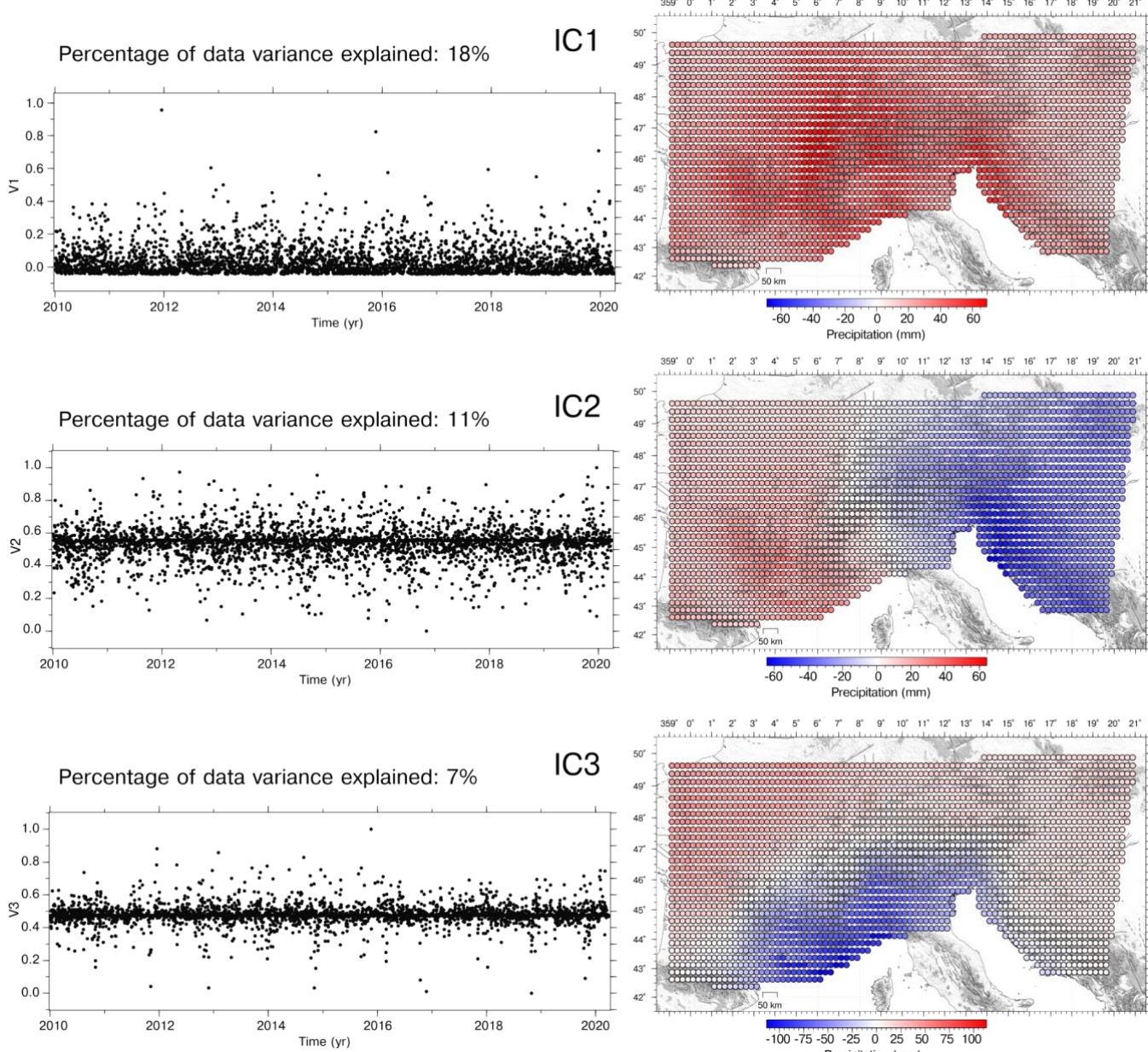

Figure 14: IC1, IC2 and IC3 of the RAIN decomposition.

This supports the hypothesis that precipitation, atmospheric pressure, hydrological loading and ground displacement are somehow interconnected and characterized by a common climate-related forcing, whose characteristics of spatial variability are described by the NAO and Atlantic Ridge weather regimes.

We point out that HYDL, NTAL and GNSS are models or measurements of vertical displacements, which are positive when upward and negative when downward; while RAIN is the amount of fallen rain per unit area.

Let us consider for the sake of simplicity the IC1 case, but what we are going to discuss holds true also for IC2 and IC3.
The temporal evolution of NTAL_IC1 (NTAL_V1) is correlated with the temporal evolution of RAIN_IC1 (RAIN_V1, Fig.
15g-i) and anti-correlated with the time derivative of the temporal evolution of HYDL_IC1 (HYDL_V1, Fig. 15a-c).
HYDL_V1 is also highly anti-correlated with RAIN_IC1 (Fig. 15d-f).

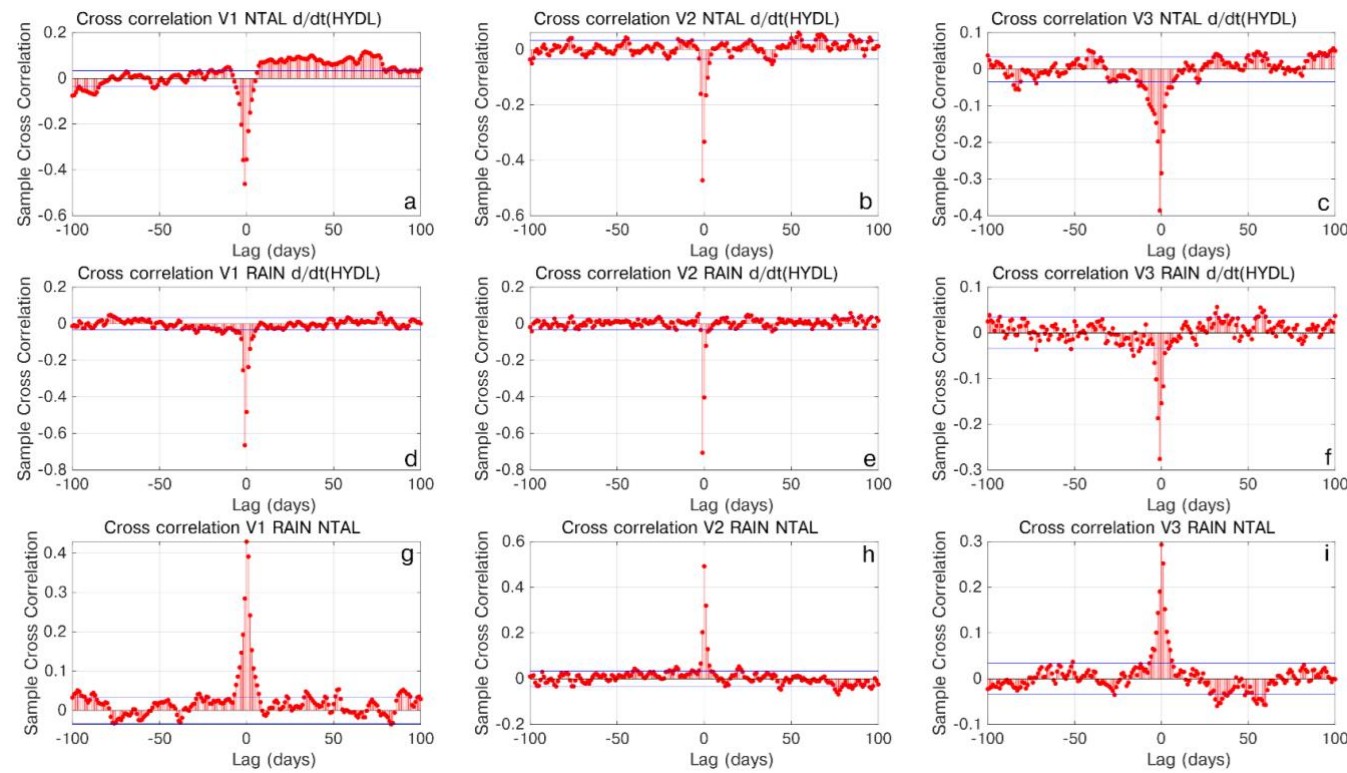


**Figure 15: Cross correlation between:**
**a) the temporal evolution of the IC1 of the NTAL decomposition and the time derivative of the temporal evolution of the IC1 obtained**
**by decomposing HYDL; b) same as a), but considering IC2; c) same as a), but considering IC3;**
**d) the temporal evolution of the IC1 of the precipitation data decomposition and the time derivative of the temporal evolution of the**
**IC1 obtained by decomposing HYDL; e) same as d), but considering IC2; f) same as d), but considering IC3;**
**g) the temporal evolution of the IC1 of the NTAL decomposition and the temporal evolution of the IC1 of the precipitation data**
**decomposition; h) same as g), but considering IC2; i) same as g), but considering IC3.**

Our interpretation of the correlations discussed above, schematically represented in Fig. 16, is the following: when the weather
goes from a low pressure to a high pressure regime, the increasing pressure causes a downward displacement of the ground
(Fig. S8). Anyway, low pressure regimes are often associated with precipitation, and that is why IC1_RAIN and IC1_NTAL
are correlated. It follows that when we go from high pressure to low pressure conditions, the ground motion, if we assume a
pure elastic process, is affected by two forces acting in opposite directions: the decreasing atmospheric pressure induces uplift,

while the precipitation load causes downward motion. Rain also affects hydrological loading, increasing it and causing a downward ground motion. As a consequence, the temporal derivative of HYDL_IC1, which is more sensitive to small but fast variation of hydrological loading than HYDL itself, is negative and anti-correlated with IC1_RAIN.

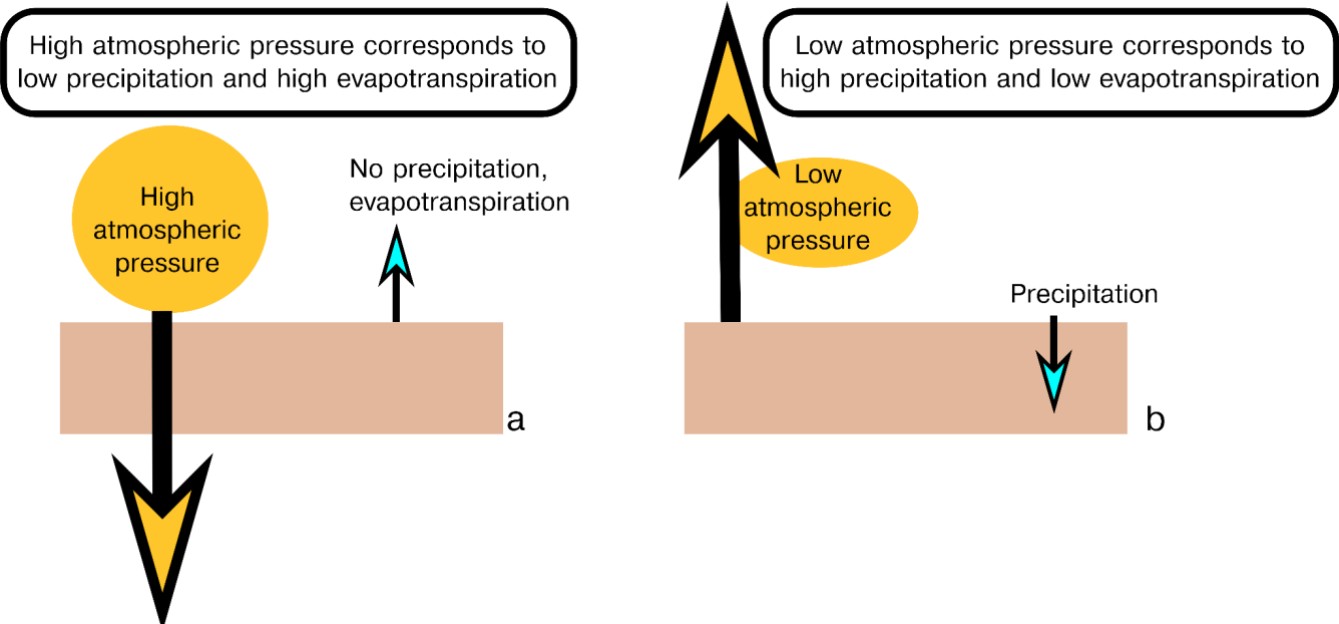

**Figure 16: Schematic representation of the ground vertical displacement due to elastic deformation during high pressure (a) and low pressure (b) conditions. Yellow arrows reflect displacements associated with atmospheric pressure, blue arrows reflect displacements associated with precipitation and evapotranspiration.**

Atmospheric pressure variations happen at fast temporal scales, then the switch from high to low pressure conditions (and vice versa) can happen in a few days and cause quite large (centimetric) ground vertical displacements. Hydrological loading acts at longer timescales and there are several factors to consider besides precipitation, in particular the temperature, which causes evapotranspiration. Nonetheless, computing the time derivative of the hydrological loading allows to detect "fast" variations due to the change of the atmospheric pressure and the precipitation events often associated with it.

The interpretation of IC4 is less straightforward and the pattern we see in the Alps (Figure S.15) is not easy to explain. Air temperature increase can induce both positive and negative vertical displacements. One possible mechanism to explain negative vertical displacements associated with temperature increase is that in the alpine valleys the water content increases as the temperature increases because of the snow and ice melting. It follows that in those areas the elastic response to hydrological load is higher during summertime than winter, as observed by Capodaglio et al. (2017), so that negative vertical displacements are measured when the temperature increases. Then, it is not surprising that in the alpine valleys the stations affected by large IC4-related displacements move downward as temperature increases. This may be an example of a small-

scale hydrological process that is likely badly reproduced by the HYDL displacement dataset, which does not have a spatial resolution fine enough to represent hydrological loading displacements at the scale of the alpine valleys. Other site-dependent processes that can potentially induce uplift during winter are the ice formation, and subsequent melting, in the antenna and antenna mount (Koulali and Clarke, 2020) and soil freezing (Beck et al., 2015).

Conversely, positive vertical displacements as the temperature increases can be caused by monument/bedrock thermal expansion and the drying of the soil, because of the reduction of the hydrological load. While HYDL takes into account the drying of the soil, we cannot exclude that some local, unmodeled, environmental conditions can amplify this effect at some sites. This might explain why most of the sites affected by uplift during temperature increases are located in plain areas, like the northern sector of the Paris Basin and in the Po plain, instead of the mountainous ones. The relation between IC4 and local processes is also suggested by the heterogeneity of this signal in terms of its spatial distribution, sign, amplitude and relevance in explaining the data variance. In fact, while ~50% of the stations have U4<2mm (Fig. S3d) and explain <1% of the data variance, meaning that IC4 is almost unuseful to reproduce the original data, there is a non-negligible number of stations (~10%) explaining >10% of the data variance and with U4>6mm. Finally, possible sources of this seasonal signal might be systematic errors in GNSS observations and in their modeling (Chanard et al., 2020). In the introduction we mentioned the effects of the non-tidal ocean loading on the vertical displacements and both LSDM-based and EOST models provide estimation of them. In the study region, this process induces displacements that are significantly smaller than both atmospheric and hydrological loading, due to the distance from the oceans of the study area, so we do not take it into account. According to the estimation of the LSDM-based model, the maximum amplitude of the spatial mean over the study region of the displacements associated with it is 4.3 mm; while the maximum amplitude of the displacements associated with atmospheric and hydrological loading are 23.8 mm and 12.2 mm, respectively. Figure S5 provides a comparison of the spatial mean of the displacements associated with the three deformation mechanisms.

## 5.3 Vertical velocity gradients across the Alps

The vertical velocity field of the IGb14-time series and of the IGb14-time series with the contribution of the first 4 ICs removed (ICs filtered) do not differ much in terms of uplift/subsidence patterns (see Fig. 11), both showing the belt of continuous uplift, of the order of 1-2 mm yr$^{-1}$, along the Alpine mountain chain. As shown in Fig. 11c, the vertical velocities from filtered time-series show barely faster positive rates, mainly as an effect of filtering out hydrological and atmospheric displacements of IC1, as discussed above. Figure 17 shows the continuous vertical velocity field obtained from the discrete values adopting the multiscale, wavelet-based, approach described in Tape et al. (2009), and some vertical velocity and topographic profiles running across the great Alpine area. The same figure obtained using velocities and uncertainties from unfiltered time-series is shown in the Supplementary Information (Fig. S20). Despite the similarity in the velocity patterns, the improvements in both the precision and consistencies of vertical spatial gradients are apparent in cross section view. Profile E-E' in Fig. 17 shows positive vertical rates increasing from W to E, with the maximum uplift rates in the central Alps, and the positive correlation with the topography along the chain axis, with decreasing rates toward the east, changing to subsidence east of

Lon. ~14.5° E, while entering the Pannonian basin domain. The correlation with topography is also clear in the chain-normal profiles (A-A', B-B', C-C' and D-D'). In the Western and Central Alps (A-A' and B-B') the maximum uplift rates are located in correspondence with the maximum elevation, whereas in the Eastern Alps (C-C' and D-D') the maximum uplift rates are shifted southward. The Eastern Southern Alps is the region where the largest part of the Adria-Eurasia converge is accommodated (1-3 mm yr$^{-1}$), through active thrust faults and shortening (Serpelloni et al., 2016). Here, maximum uplift rates are likely due to interseismic deformation, and their position, across the belt, is driven by thrust fault geometries, slip-rates and locking depths (Anderlini et al., 2020). Concerning the south Alpine foreland in the Po Plain and Venetian plain, Fig. 17 shows a decrease in the vertical velocities from west to east, with barely positive rates in the western Po Plain and increasing subsidence rates in the northern Adriatic and in the northern Apennines foreland.

In the Alpine foreland, positive, sub-mm yr$^{-1}$, velocities are present in the Jura Mts. and the Molasse basin, but uplift extends further northward in the Black Forest and the Franconian Platform, in southern Germany, and in the southern part of the Bohemian Massif. Overall, in the portion of central Europe investigated in this work, we see two different patterns: prevalent stable to slowly-subsiding sites (< 1 mm yr$^{-1}$) are present west of the Rhine graben, whereas a prevalence of slowly uplifting sites (< 1 mm yr$^{-1}$) is present east of it. Profile F-F' in Fig. 17 better highlights this pattern. Across the Upper Rhine Graben, the weak uplift signal in the graben's shoulders, the Vosges Mts and Black Forest, is associated with subsidence of stations located within the graben, according to Henrion et al. (2020). To the east, uplift in the Franconian Platform and the Bohemian Massif is only partially correlated with topography. It is still debated whether uplifted regions across NW Europe attest to lithospheric buckling in front of the Alpine arc or were randomly produced by a swarm of baby plumes. Uplift propagation by interferences with the Western Carpathians and possible mantle processes, as suggested by the positive dynamic and residual topography (Faccenna et al., 2014), may contribute to the observed uplift in the Bohemian Massif.

Sternai et al. (2019) investigated the possible relative contribution of different geophysical and geological processes in the actual vertical velocity budget over the Alps, suggesting that the interaction among tectonic and surface mass redistribution processes, rather than an individual forcing, better explain vertical deformation in the Alps. Mey et al. (2016) suggested that ~90% of the present-day uplift of the Alpine belt is due to the melting of the LGM ice cap. While it is difficult to independently constrain the patterns and magnitude of mantle contributions to ongoing Alpine vertical displacements at present, lithospheric adjustment to deglaciation and erosion are by far the most important ongoing process, but other authors suggest that other processes are currently shaping the vertical ground motion pattern. In the western and central Alps, active convergence is inactive or limited, the residual uplift rates, after correction from isostatic contributions, are likely due to deep-seated mantle processes, including for example detachment of the western European slab and dynamic contributions related to sub-lithospheric mantle flow (Chery et al., 2016; Nocquet et al., 2016; Sternai et al., 2019). A tectonic contribution to the ongoing uplift is, instead, more likely in the Eastern Alps, and in particular in the Southeastern Alps, where the Adria-Europe convergence is accommodated. However, Anderlini et al (2020) observed that more accurate glacio isostatic models would be needed when interpreting tectonic contributions to uplift at the edge of ice caps, as in the Eastern Southern Alps.

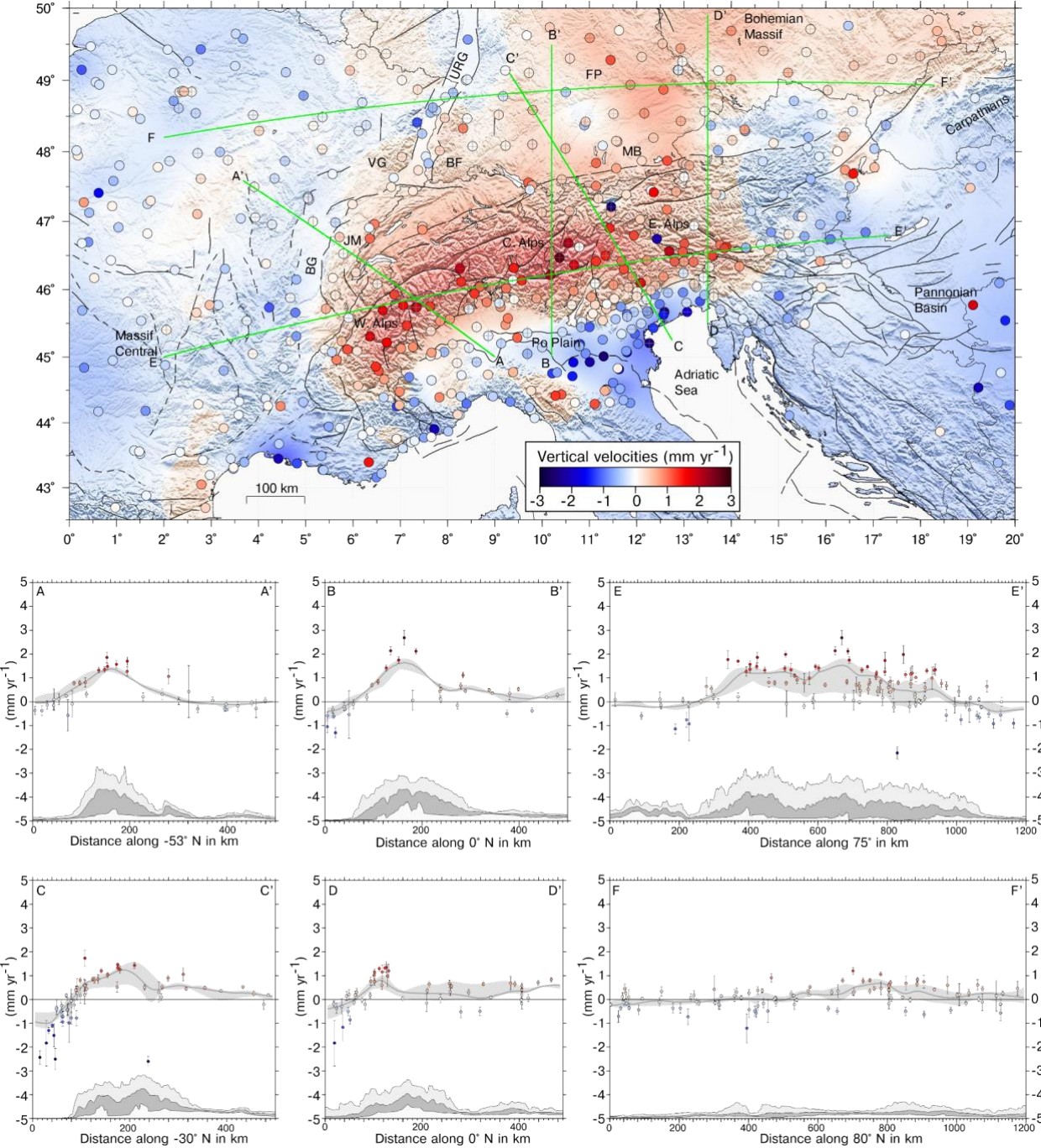

**Figure 17: Vertical velocities from filtered time-series (colored circles), continuous velocity field, topographic and swath profiles across the great Alpine area. Each profile (green line) encompasses a 50+50 km swath. BG: Bresse Graben; JM: Jura Mts.; VG: Vosges Mts.; BF: Black Forest; URG: Upper Rhine Graben; FP: Franconian Platform; MB: Molasse Basin.**

## 6 Conclusions

The application of a blind source separation algorithm to vertical displacement time-series obtained from a network of GNSS stations in the Great Alpine Area allows us to identify the main sources of vertical ground deformation. Besides the linear trend, vertical displacements are influenced by: 1) atmospheric pressure loading, 2) hydrological loading and 3) seasonal processes in phase with temperature. The analysis of displacement time series of environmental loading shows that the largest vertical motions are related to the variation of atmospheric pressure, in particular when considering daily/weekly timescales. Seasonal displacements are more clearly associated with hydrological loading and processes in phase with temperature. However, while deformation associated with temperature is well isolated, we were not able to clearly separate the atmospheric and hydrological loading signals in the GNSS displacement time-series.

We use the results of the time-series decomposition to filter the IGb14 time-series and study the effect of removing signals associated with environmental loading and temperature-related processes on the vertical velocities and uncertainties. Removing these signals causes a quite uniform, but limited (~0.1 mm yr$^{-1}$), increase of the velocities, which we interpret as due to the small negative linear trend associated with the atmospheric and hydrological loading-induced displacements. It is worth noting that the procedure used in this work to estimate the station velocities does not allow to distinguish the tectonic velocities from the contribution to the velocity induced by climate-related processes, in particular if the linear trend associated with ATML and/or HYDL time series is large. Furthermore, the filtering almost halves the uncertainties associated with the velocities and changes the noise spectra, increasing the white noise percentage to the detriment of the colored one.

Although providing a geological/geophysical explanation for the observed vertical velocity pattern is out of the scope of this work, we can conclude that more precise and accurate vertical velocities, such as the one presented in this work, can be obtained by careful signal detection and filtering. This can help develop better spatially resolved models, aiming at a more effective understanding of the relative contribution of the different ongoing geodynamic and tectonic processes shaping the present-day topography of the Alps.

**Code and data availability**

The MATLAB code for vbICA decomposition is available from http://dx.doi.org/10.17632/n92vwbg8zt.1. Global datasets used for the hydrological, atmospheric and ocean load model are taken from http://loading.u-strasbg.fr/ (EOST model) and http://rz-vm115.gfz-potsdam.de:8080/repository/entry/show?entryid=24aacdfe-f9b0-43b7-b4c4-bdbe51b6671b (LSDM-based model).Precipitation data are available on https://disc.gsfc.nasa.gov/datasets/GPM_3IMERGDF_06/summary. Temperature data are available on https://www.ecad.eu/download/ensembles/download.php and IGb14 GPS time series on https://doi.pangaea.de/10.1594/PANGAEA.938422.

## Author contribution

F. Pintori conceived and led the paper, E. Serpelloni coordinated the study and analyzed GNSS data, A. Gualandi supervised the vbICA analysis of GNSS displacements. All the authors discussed the content of the paper and shared the writing.

## Competing interests

The authors declare that they have no conflict of interest.

## Acknowledgements

We thank E. Scoccimarro and M. Zampieri for fruitful suggestions on the interpretation of meteo-climatic data. F. Pintori was supported by the project TRANSIENTI, founded by the Italian Ministry of Education, Universities and Research (MIUR) "Premiale 2014". Adriano Gualandi is supported by European Research Council Advance Grant 835012 (TECTONIC). This work has been developed in the framework of the project KINDLE, funded by the "Pianeta Dinamico" INGV institutional project. We acknowledge the E-OBS dataset from the EU-FP6 project UERRA https://www.uerra.eu) and the Copernicus Climate Change Service, and the data providers in the ECA&D project (https://www.ecad.eu). We are grateful to the many agencies, companies and networks that have made GNSS data available. We specifically thank the following public networks and institutions for raw RINEX data: IGS, EUREF-EPN, AGROS (Serbia), CZEPOS (Czech Republic), GPS-EMILIA ROMAGNA (Italy), InOGS-FREDNET (Italy), Rete GNSS Marussi FVG (Italy), ASI-GEODAF (Italy), GEONAS (Czech Republic), GFZ (Germany), GREF (Germany), Leica-Geosystem HXGN-SmartNeT (Italy), GNSS LIGURIA (Italy), Topcon Positioning Italy NETGEO (Italy), OLGGPS (Austria), RENAG (France), RGP (France), INGV-RING (Italy), SIGNAL (Slovenia), SONEL, SPINGNSS (Italy), STPOS (BZ, Italy), TPOS (TN, Italy), GPS-VENETO (Italy), VESOG (Czech Republic). ORPHEON data were provided to the authors for scientific use in the framework of the GEODATA-INSU-CNRS convention. We acknowledge Echtzeit Positionierung Austria for providing access to the EPOSA data. SAPOS networks are operated by various German States (Landesamt für Digitalisierung, Breitband und Vermessung and Baden-Württemberg).

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
