# Peer review of "Common mode signals and vertical velocities in the great Alpine area 1"

_Solid Earth, 2021_

## Referee Comment (RC1)

Pintori et al. 2021

The manuscript "Common mode signals and vertical velocities in the great Alpine area from GNSS data" by Pintori et al. 2021 utilizes variational Bayesian Independent Component Analysis to separate loading signals due to atmospheric and hydrologic sources from vertical GNSS time series to further isolate uplift rates due to mountain building in the greater alpine region. This also serves to provide additional information on the source of the common mode signals in GNSS time series. In this study, they also compare the vbICA method to other methods of constructing the common mode. Finally, they remove the non-mountain building signals from the data and estimate uplift rates. Overall, the article is well written and is a useful contribution. However, there are a few areas that would benefit from additional clarification and added detail.

**General comments:**

*ICA method in general*
To my understanding, an ICA is meant to separate the different sources out of a given signal. This leads me to interpret that each component would thus represent a different source. However, throughout the paper it seems to me that each component is not necessarily a singular source (eg Figure 2). This component is obviously due to some combination of tectonic trend as well as some seasonal signal likely due to hydrology or atmospheric loading. If the point of using an ICA is to separate out different sources so that you can further isolate a specific source, how can you be sure that you are fully capturing the signal you think you are (in this case the tectonic signal or later on NTAL/HYDL)?

To this end, why are you decomposing the "source" signals (NTAL and HYDL)? Wouldn't a component from the GNSS decomposition represent the NTAL signal? Or the HYDL signal? And then why do you combine NTAL IC1 and HYDL IC1 and compare them to the GNSS IC1? This implies this component is a portion (and only a portion) of two very different sources. How do you know that's all that's in there? I suppose, what I'm asking is some further clarification in the text about (1) what the different components actually mean in terms of "sources" insofar as are they "sources"? or just spatially independent signals/temporally independent and thus could be heavily influenced by certain things but not necessarily the entire signal (2) further explanation for the motivation behind decomposing the source signals (NTAL/HYDL) and why it's necessary. I realize some of this is not specific to this paper but ICA in these applications in general but I think the text would greatly benefit from further explanation.

*Decomposition of the NTAL/HYDL*
The spatial pattern of the different components from the NTAL and HYDL are incredibly similar. Is this due to how the algorithm works or are these signals just by coincidence showing very similar spatial patterns. How much of the variance due each of these components represent? I think including that, maybe even just in the figures would be helpful for interpretation of the different components.

*Temperature*
I agree that the fourth component is well correlated with temperature. However, temperature is just a strong seasonal signal so couldn't this signal be something else? In lines 369-370, you mention that when temperature increases the stations in the mountains subside. I'm just confused by what physical mechanism would cause this. The two mechanisms that you list for temperature in lines 505, don't explain why the mountains would experience downward deflection during warm periods. Can you provide further explanation for the physical cause of this? I think in the paper you indicate too heavily that this component is due to temperature fluctuations (especially in Figure 8 and the associated text, the conclusion and abstract) and don't necessarily support this. Correlation does not always indicate causation. I think further data and text is needed to support this finding. Especially since this is mentioned in the abstract (line 16) as well as the conclusion (line 586/593).

*Application of vbICA for removing NTAL and HYDL*
Martens et al 2020 (J. of Geodesy) highlighted the importance of removing NTAL and NTOL signals from GNSS timeseries to reduce scatter/dispersion. In lines 351-353, you mention, vbICA may not be able to separate the NTAL vs HYDL signals. Why not just remove the signals using the GFZ products instead of using the ICA method? Does removing the ICA reduce the scatter more than just removing the signals to begin with? -

**Minor comments:**

GNSS processing - Do remove signals due to earthquakes? In the supplement you mention removing offsets due to equipment changes but don't mention offsets or post seismic signal removal. Does the ICA capture earthquake signals? Wouldn't this be a good signal to remove to better isolate the uplift?

Line 123: grammatical issue - "Since they allow to account"
Line 254: grammatical issue – "its temporal evolution has not a domination frequency"

Lines 230: What reference frames are you using for the NTAL and HYDL models?

Lines 250-251: There are no units for y-axes on the temporal portion of the components. What are the units? Are there any? To construct the signal at a given spot, do you multiply the temporal by the spatial displacement for that point? It would be helpful for understanding the figures.

Lines 340: For the second component, you list Pearson's correlation coefficient in addition to the Lin's. Can you list the Pearson's for component 1? And in the third component, is this the Pearson's or the Lin's coefficient?

Line 338: How many stations have displacements above 3mm?

Line 389: Is the k value -2 for both?

Line 404-408: I think there's a typo here.

*Consequently, the **unfiltered** time series are modeled only with the linear trend plus the temporal correlated noise, while the **unfiltered** time series modeling annual and semi-annual terms are also included.*

Are both unfiltered? I think the first one should be filtered, yes?

Section 5.3: If you are removing the linear trend, then are your uplift rates non-tectonic uplift? Or are you adding that back in? Just confusing since in the introduction it seemed like you were settling up to better estimate uplift rates due to tectonics? I think it's fine to remove the linear trend for comparison of stacking methods ect but for Figure 13 and discussion in 5.3 is this with the linear trend removed or included? I the nontectonic uplift? Or are you adding the linear trend back in? Can you clarify?

Many of the figures appear blurry. Additionally, the font on the axes of many of the figures is incredibly difficult to read (eg Figure 3) and would benefit from larger font size.

---

## Author Comment (AC1)

**General comments:**

**ICA method in general**

**To my understanding, an ICA is meant to separate the different sources out of a given signal. This leads me to interpret that each component would thus represent a different source. However, throughout the paper it seems to me that each component is not necessarily a singular source (eg Figure 2). This component is obviously due to some combination of tectonic trend as well as some seasonal signal likely due to hydrology or atmospheric loading. If the point of using an ICA is to separate out different sources so that you can further isolate a specific source, how can you be sure that you are fully capturing the signal you think you are (in this case the tectonic signal or later on NTAL/HYDL)?**

We are aware that vbICA may not properly separate the tectonic, linear, trend from seasonal non-tectonic signals (Figure 2), as already highlighted by Gualandi et al. (2016). In fact, when the linear trend is not removed from the time series, they result to be highly correlated and this prevents the vbICA algorithm from working efficiently. That is why we removed the linear trend from the time series. We have accomplished this task in a multivariate sense, rather than using standard trajectory models. Nonetheless, even using detrended time series as vbICA input, we cannot separate the NTAL from the HYDL contribution. Considering that they are both a consequence of meteoclimatic forcings, they maintain some interdependence. As a consequence, while NTAL and HYDL are different loading types, they might not be independent from a mathematical point of view and so vbICA is not able to properly separate them. In order to explain that, we add the following part in line 503:

*"The vbICA algorithm is not able to separate NTAL and HYDL because they are not independent from a mathematical point of view. This emerges also from the recent work by Tan et al. (2022), who performed an ICA on GNSS time series of the Yunnan Province of China and interpreted IC1 as the average effects of the joint patterns from soil moisture and atmospheric-induced annual surface deformations. Let us consider for example the case of IC2_NTAL and IC2_HYDL. They have two different temporal evolutions (V2_NTAL and V2_HYDL); but the spatial distributions (U2_NTAL and U2_HYDL) have the same pattern, i.e. they only differ for a weighting factor k. Then, we can write U2_NTAL=k\*U2_HYDL.*

*The displacement d resulting from the combined effect of IC2_NTAL and IC2_HYDL is then:*

*d= IC2_NTAL + IC2_HYDL= U2_NTAL\*V2_NTAL + U2_HYDL\*V2_HYDL= U2_HYDL\*(k\*V2_NTAL+V2_HYDL).*

*As a result, the displacement due to IC2_NTAL + IC2_HYDL is identified by a single spatial distribution U2_HYDL and a temporal evolution k\*V2_NTAL+V2_HYDL. Then, if we do not make any prior assumptions about V2_NTAL and V2_HYDL, it is not possible to separate IC2_NTAL and IC2_HYDL from a statistical point of view.*

*In Sect. 4.2 we show that not only IC2_NTAL and IC2_HYDL have very similar spatial patterns, but also IC1_NTAL and IC1_HYDL, IC3_NTAL and IC3_HYDL have similar spatial responses. Then, the GNSS time-series decomposition in the Alpine area does not allow separating the effect of the hydrological loading from the atmospheric loading with an ICA approach."*

**To this end, why are you decomposing the "source" signals (NTAL and HYDL)? Wouldn't a component from the GNSS decomposition represent the NTAL signal? Or the HYDL signal? And then why do you combine NTAL IC1 and HYDL IC1 and compare them to the GNSS IC1? This implies this component is a portion (and only a portion) of two very different sources. How do you know that's all that's in there? I suppose, what I'm asking is some further clarification in the text about (1) what the different components actually mean in terms of "sources" insofar as are they "sources"? or just spatially independent signals/temporally independent and thus could be heavily influenced by certain things but not necessarily the entire signal (2) further explanation for the motivation behind decomposing the source signals (NTAL/HYDL) and why it's necessary. I realize some of this is not specific to this paper but ICA in these applications in general but I think the text would greatly benefit from further explanation.**

We don't decompose the HYDL and NTAL datasets with vbICA with the goal to separate different sources, but in order to investigate the presence of any spatiotemporal signatures, like the ones emerging from the vbICA analysis of the GNSS data (IC1, IC2 and IC3), that could help in the interpretations of the ICA decomposition.

At the beginning of Section 4.2 we have added some text to further explain the reason why we decompose NTAL and HYDL and what the different components mean in terms of "sources":

*"As discussed in the introduction, atmospheric and hydrological loading are likely the main sources of vertical displacement in the great Alpine region. Since they are both uniform in terms of spatial response, showing smooth spatial variations, we decided to check if the first 3 ICs of the GNSS decomposition are associated with the displacements due to atmospheric and hydrological loading, and with their pattern of variability.*

*The vbICA analysis separates the data into statistically independent signals, which is useful because independent signals are often caused by different and independent sources of deformation. Nonetheless, a single source of deformation, such as atmospheric or hydrological loading, can be spatially heterogeneous and characterized by peculiar spatio-temporal patterns. In this case, the vbICA separates a single source of deformation in different components associated with different spatio-temporal patterns. As a consequence, we decided to apply a vbICA decomposition on HYDL and NTAL model displacement time series in order to check if they show any pattern and if they resemble the spatial distribution of IC1, IC2 and IC3 of the GNSS decomposition."*

We also decide to change Figure 7, showing the displacement of two different sites, one located in the south-western part of the study region (STV2), the other in the north-eastern side (LYSH), so that the displacement associated with GNSS_IC2 and GNSS_IC3 have opposite sign. This figure show that both the amplitude and the temporal evolution of the GNSS and HYDL+NTAL signals are quite similar in all the three components: while we might have not recognized other sources of deformation generating those signals, atmospheric and hydrological loading seem by far the most relevant processes causing the displacements associated with GNSS_IC1, GNSS_IC2 and GNSS_IC3.

[Figure]

*Figure 7: Comparison, at the LYSH (Lon: 18.45°; Lat: 49.55°) site, between the displacements associated with: a) GNSS_IC1 and NTAL+HYDL_IC1; b) GNSS_IC2 and NTAL+HYDL_IC2; c) GNSS_IC3 and NTAL+HYDL_IC3. d), e), f) are the same as a), b), c), respectively, for the STV2 (Lon: 6.11°; Lat: 44.57°) site. A 30-days moving average filter is applied to better visualize the data.*

**Decomposition of the NTAL/HYDL**

**The spatial pattern of the different components from the NTAL and HYDL are incredibly similar. Is this due to how the algorithm works or are these signals just by coincidence showing very similar spatial patterns. How much of the variance due each of these components represent? I think including that, maybe even just in the figures would be helpful for interpretation of the different components.**

The similarities between the spatial patterns of the NTAL and HYDL independent components do not depend on how the algorithm works: the relative position of the sites is never taken into account during the analysis.

The presence of N-S and E-W gradients in the ICs of both NTAL and HYDL is caused by their link to a common, meteo-climatic, source. In fact, atmospheric and hydrological loading depend on the climatic conditions, which are spatially and temporally variable.

In section 5.2 we add details about the interconnection between precipitation, atmospheric pressure and hydrological loading:

"*We also performed a vbICA analysis on precipitation data (RAIN) recorded over the study region, using 3 ICs. The spatial pattern of the ICs is analogous to the ones associated with NTAL and HYDL (Fig. 14).*

[Figure]

*Figure 14: IC1, IC2 and IC3 of the RAIN decomposition.*

This supports the hypothesis that precipitation, atmospheric pressure, hydrological loading and ground displacement are somehow interconnected and characterized by a common climate-related forcing, whose characteristics of spatial variability are described by the NAO and Atlantic Ridge weather regimes.

We point out that HYDL, NTAL and GNSS are models or measurements of vertical displacement, which is positive when upward and negative when downward; while RAIN is the amount of fallen rain per unit area.

Let us consider for the sake of simplicity the IC1 case, but what we are going to discuss holds true also for IC2 and IC3.

The temporal evolution of NTAL_IC1 (NTAL_V1) is correlated with the temporal evolution of RAIN_IC1 (RAIN_V1, Fig. 15g-i) and anti-correlated with the time derivative of the temporal evolution of HYDL_IC1 (HYDL_V1, Fig. 15a-c). HYDL_V1 is also highly anti-correlated with RAIN_IC1 (Fig. 15d-f).

[Figure]

*Figure 15: Cross correlation between:*
*a) the temporal evolution of the IC1 of the NTAL decomposition and the time derivative of the temporal evolution of the IC1 obtained by decomposing HYDL; b) same as a), but considering IC2; c) same as a), but considering IC3;*
*d) the temporal evolution of the IC1 of the precipitation data decomposition and the time derivative of the temporal evolution of the IC1 obtained by decomposing HYDL; e) same as d), but considering IC2; f) same as d), but considering IC3;*
*g) the temporal evolution of the IC1 of the NTAL decomposition and the temporal evolution of the IC1 of the precipitation data decomposition; h) same as g), but considering IC2; i) same as g), but considering IC3.*

*Our interpretation of the correlations discussed above, schematically represented in Fig. 16 is the following: when the weather goes from a low pressure to a high pressure regime, the increasing pressure causes a downward displacement of the ground (Fig. S8). Anyway, low pressure regimes are often associated with precipitation, and that is why IC1_RAIN and IC1_NTAL are correlated. It follows that when we go from high pressure to low pressure conditions, the ground motion, if we assume a pure elastic process, is affected by two forces acting in opposite directions: the decreasing atmospheric pressure induces uplift, while the precipitation load causes downward motion. Rain also affects hydrological loading, increasing it and causing a downward ground motion. As a consequence, the temporal derivative of HYDL_IC1, which is more sensitive to small but fast variation of hydrological loading than HYDL itself, is negative and anti-correlated with IC1_RAIN.*

[Figure]

*Figure 16: Schematic representation of the ground vertical displacement due to elastic deformation during high pressure (a) and low pressure (b) conditions. Yellow arrows reflect displacements associated with atmospheric pressure, blue arrows reflect displacements associated with precipitation and evapotranspiration.*

*Atmospheric pressure variations happen at fast temporal scales, then the switch from high to low pressure conditions (and vice versa) can happen in a few days and cause quite large (centimetric) ground vertical displacements. Hydrological loading acts at longer timescales and there are several factors to consider besides precipitation, in particular the temperature, which causes evapotranspiration. Nonetheless, computing the time derivative of the hydrological loading allows to detect "fast" variations due to the change of the atmospheric pressure and the precipitation events often associated with it."*

Besides IC1, which is a spatially uniform signal explaining more than the 90% of the total variance in either NTAL or HYDL decomposition, IC2 and IC3 probably reveals the spatio-temporal features of the weather regimes that cause atmospheric and hydrological loading on the surface: the Atlantic Ridge and the North Atlantic Oscillation. In section 5.2 we added the following part:

*"It is then likely that weather regimes like the NAO and the Atlantic Ridge influence both NTAL and HYDL, which is mainly forced by precipitation, so that the spatial patterns of the ICs associated with atmospheric and hydrological loading are the same of NAO (N-S) and Atlantic Ridge (E-W)."*

The percentage of total variance explained by each component is added to the figures.

**Temperature**

**I agree that the fourth component is well correlated with temperature. However, temperature is just a strong seasonal signal so couldn't this signal be something else? In lines 369-370, you mention that when temperature increases the stations in the mountains subside. I'm just confused by what physical mechanism would cause this. The two mechanisms that you list for temperature in lines 505, don't explain why the mountains would experience downward**

**deflection during warm periods. Can you provide further explanation for the physical cause of this? I think in the paper you indicate too heavily that this component is due to temperature fluctuations (especially in Figure 8 and the associated text, the conclusion and abstract) and don't necessarily support this. Correlation does not always indicate causation. I think further data and text is needed to support this finding. Especially since this is mentioned in the abstract (line 16) as well as the conclusion (line 586/593).**

Thanks for this comment. We agree that the conclusions on IC4 are too strong. We modify the abstract, changing a sentence that incorrectly lets the reader suppose that temperature might directly cause the displacement associated with IC4.

In fact, it is more correct to state that the displacements associated with IC4 are caused by processes correlated to temperature, which are discussed in section 5.2, than caused by temperature itself. The contrasting behavior observed for some stations in the Alps and the Adriatic foreland is difficult to explain. Here we propose a possible mechanism, that is now, hopefully, clearly described in Section 5.2 (from line 504):

*"Air temperature increase can induce both positive and negative vertical displacements. In the alpine valleys the water content increases as the temperature increases because of the snow and ice melting. It follows that in those areas the elastic response to hydrological load is higher during summertime than winter, as observed by Capodaglio et al., (2017), so that negative vertical displacements are measured when the temperature increases. Then, it is not surprising that in the alpine valleys the stations affected by large IC4-related displacements move downward as temperature increases. This may be an example of a small-scale hydrological process that is likely badly reproduced by the HYDL displacement dataset, which does not have a spatial resolution fine enough to represent hydrological loading displacements at the scale of the alpine valleys. Other site-dependent processes that can potentially induce uplift during winter are the ice formation, and subsequent melting, in the antenna and antenna mount (Koulali and Clarke, 2020) and soil freezing (Beck et al., 2015).*

*Conversely, positive vertical displacements as the temperature increases can be caused by monument/bedrock thermal expansion and the drying of the soil, because of the reduction of the hydrological load. While HYDL takes into account the drying of the soil, we cannot exclude that some local, unmodeled, environmental conditions can amplify this effect at some sites. This might explain why most of the sites affected by uplift during temperature increases are located in plain areas, like the northern sector of the Paris Basin and in the Po plain, instead of the mountainous ones.*

*The relation between IC4 and local processes is also suggested by the heterogeneity of this signal in terms of its spatial distribution, sign, amplitude and relevance in explaining the data variance. In fact, while ~50% of the stations have U4<2mm (Fig. S3d) and explain <1% of the data variance, meaning that IC4 is almost unuseful to reproduce the original data, there is a non-negligible number of stations (~10%) explaining >10% of the data variance and with U4>6mm."*

**Application of vbICA for removing NTAL and HYDL**

**Martens et al 2020 (J. of Geodesy) highlighted the importance of removing NTAL and NTOL signals from GNSS time series to reduce scatter/dispersion. In lines 351-353, you mention, vbICA may not be able to separate the NTAL vs HYDL signals. Why not just remove the signals using the GFZ products instead of using the ICA method? Does removing the ICA reduce the scatter more than just removing the signals to begin with?**

Our goal is to remove signals associated with meteo-climatic processes using vbICA, instead of subtracting modeled displacements, such as those made available through loading services like GFZ, from the measured displacements. This approach minimizes biases due to the mismatch between the actual signal caused by atmospheric and hydrological loading and the modeled ones. Larochelle et al. (2018) reached similar conclusions by comparing GRACE measurements and the results from ICA decompositions of GNSS displacements, which resulted to be more accurate in correcting GNSS from seasonal displacements than removing GRACE displacements, which smooth local effects in the data acquisition and processing.

This is now described at the beginning of Section 5.1:

*"Our goal is to estimate the tectonic velocity of the GNSS stations, then we seek to remove signals associated with meteo-climatic processes. Instead of subtracting from the IGb14-time series the modeled displacements, such as those made available through loading services like GFZ, we prefer to subtract the displacements associated with the ICs. This approach minimizes biases due to the mismatch between the actual signal caused by atmospheric and hydrological loading and the modeled ones. Larochelle et al. (2018) reached similar conclusions by comparing GRACE measurements and the results from ICA decompositions of GNSS displacements, which resulted to be more accurate in correcting GNSS from seasonal displacements than removing GRACE displacements, which smooth local effects in the data acquisition and processing.*

*In order to support the approach followed, we estimated the scatter of the GNSS displacement time series by computing the mean standard deviation of 1) the time series given as input to vbICA (IGb14-time series), 2) the IGb14-time series minus the combined displacement associated with the first 3 ICs and 3) the IGb14-time series minus the displacements due to HYDL+NTAL from GFZ models. The resulting standard deviation is 5.32, 4.10 and 4.73, respectively. This demonstrates that removing the displacement associated with the first four ICs is more effective in reducing the scatter than removing the HYDL+NTAL contribution."*

**Minor comments:**

**GNSS processing - Do remove signals due to earthquakes? In the supplement you mention removing offsets due to equipment changes but don't mention offsets or post seismic signal removal. Does the ICA capture earthquake signals? Wouldn't this be a good signal to remove to better isolate the uplift?**

Yes, we remove both instrumental and co-seismic offsets and eventually post-seismic signals. However, no co-seismic offsets interest the GNSS stations considered in this work.

**Line 123: grammatical issue - "Since they allow to account"**

Ok, now it is "allow to take into account".

**Line 254: grammatical issue – "its temporal evolution has not a domination frequency"**

Ok, we use "dominant".

**Lines 230: What reference frames are you using for the NTAL and HYDL models?**

Center of figure. We changed the text accordingly.

**Lines 250-251: There are no units for y-axes on the temporal portion of the components. What are the units? Are there any? To construct the signal at a given spot, do you multiply the temporal by the spatial displacement for that point? It would be helpful for understanding the figures.**

In order to answer these questions we added, from line 209, a more detailed explanation on how to interpret the temporal evolution, the spatial distribution and the displacement associated with the ICs.

*"Before discussing the vbICA results, we briefly explain how to interpret the temporal evolution and the spatial distribution of the ICs, so that it is possible to retrieve the displacements associated with them.*

*The color of each GNSS site in Fig. 2 represents the IC2 spatial response (U2), which indicates the maximum displacement associated with the IC2, while the temporal function V2 is normalized between 0 and 1. The displacement associated with IC2 between two epochs (e.g. $t_1$ and $t_2$, with $t_2>t_1$) at the station n is computed as $V1(t_2)*U1_n-V1(t_1)*U1_n(t_1)$, where $V1(t_2)$ is the value associated with the temporal evolution of the IC at the epoch $t_2$.*

*$U1_n$ depends on the site, but not on the epoch; its unit of measurement is mm, while V has no units of measurement. As a result, $V1*U1_n$ is in mm. It follows that if $U1_n$ is positive, as we observe for each station, and V1 is increasing ($V1(t_2)>V1(t_1)$), the stations move upward during the $t_2$-$t_1$ time interval. On the other hand, if $V1(t_2)<V1(t_1)$ the stations move downward during $t_2$-$t_1$.*

*As regards Fig. 2, assuming $t_1=2010.0$ and $t_2=2020.0$, the displacements associated with IC2 are ~30 mm upward at the "red" GNSS stations, ~30 mm downward at the "blue" GNSS stations and ~0 mm at the white ones."*

We also modify lines 255-256:

*"IC1 is a spatially uniform signal characterized by an annual temporal signature, as shown by the power spectral density (PSD) plot in Fig. 3a. The mean of the maximum amplitudes is 26 mm, while the histogram showing the distribution of displacement amplitudes is shown in Fig. S3a.*

*IC2 shows a spatial response characterized by a clear E-W gradient, but, differently from IC1, its temporal evolution has not a dominating frequency. The spatial response U2 of the eastern stations (in blue) is mainly negative, while the U2 of the western stations (in red) is mainly positive."*

**Lines 340: For the second component, you list Pearson's correlation coefficient in addition to the Lin's. Can you list the Pearson's for component 1? And in the third component, is this the Pearson's or the Lin's coefficient?**

The Pearson correlation between V1_GNSS and V1_NTAL is 0.60, while between V1_GNSS and V1_HYDL is 0.35. In the third component it is the Lin correlation.

**Line 338: How many stations have displacements above 3mm?**

IC2: 411 out of 545; IC3: 414 out of 545.

**Line 389: Is the k value -2 for both?**

Thanks for catching this, we made a mistake. Pink noise k=-1; red noise k=-2. We corrected the text.

**Line 404-408: I think there's a typo here.**

**Consequently, the unfiltered time series are modeled only with the linear trend plus the temporal correlated noise, while the unfiltered time series modeling annual and semi-annual terms are also included.**

**Are both unfiltered? I think the first one should be filtered, yes?**

      Thanks for catching this. Yes, the first one is filtered. We corrected the text.

**Section 5.3: If you are removing the linear trend, then are your uplift rates non-tectonic uplift? Or are you adding that back in? Just confusing since in the introduction it seemed like you were settling up to better estimate uplift rates due to tectonics? I think it's fine to remove the linear trend for comparison of stacking methods ect but for Figure 13 and discussion in 5.3 is this with the linear trend removed or included? Is the nontectonic uplift? Or are you adding the linear trend back in? Can you clarify?**

      Thanks for this comment that helps to make our goals more clear. In order to answer these questions and make the text more clear, we introduce a new nomenclature for the GNSS time series resulting from the analysis described in lines 169-175: IGb14-time series, which are the raw displacement time-series as obtained from the processing of GNSS data in the IGb14 reference frame, as they come from the GPS data processing (except for the correction of instrumental jumps).

      In section 5.3 we compare the IGb14-time series with the ICs filtered time series. The ICs filtered time series, as stated at lines 376-379, are the result of subtracting from the IGb14-time series the combined displacement associated with the first 4 ICs. It follows that in both IGb14-time series and ICs filtered time series the linear trend is not removed, but the linear velocities are estimated independently from the raw (IGb14-time series) and filtered time-series, and compared in terms of vertical velocities and uncertainties.

      We modified the text making that more explicit (lines 376-379).

**Many of the figures appear blurry. Additionally, the font on the axes of many of the figures is incredibly difficult to read (eg Figure 3) and would benefit from larger font size.**

      We improved the resolution and quality of the figures.

---

## Author Comment (AC2)

**Pintori et al. use a version of the ICA method (called variational Bayesian ICA) to decompose vertical GPS position time series and hydrology/atmospheric predicted loading time series around the European Alps. They study the agreement between the ICs extracted from the GPS series and from the loading models for the period from 2010 to 2020. Their main conclusions are that 1) the vertical GPS series can be separated in a tectonic linear motion and variations caused by temperature and atmospheric/hydrology loading; and that 2) improved tectonic velocities are obtained by correcting the GPS series using ICs obtained from the GPS series themselves.**

**While the volume of work is of note, especially concerning the GPS data processing, I do not think the conclusions are supported by the data and methods used by the authors. It is reasonable to say that temperature variations, atmospheric pressure variations and hydrology load variations contribute to the variations observed in vertical GPS time series, especially at the annual period, as GPS positions react to these and many other phenomena together. A completely different thing is to say that the observed GPS variations of vertical position \*are\* originated or explained by these processes, as the authors repeatedly state in the manuscript. This is a clear misinterpretation of their analysis and I develop my reasoning in the paragraphs below.**

Thanks for this comment. We are aware that GPS stations react to many processes at the same time and that the changes in the GPS positions we highlighted are the response of the solid Earth to several multiscale processes. We made the relationship between observed signals and possible causative processes less strong over the text.

**Before that, and assuming conclusion 1 is right, it's very surprising that the authors do not try to remove the modeled loadings from the GPS series to test the impact on the estimated velocities. Instead, conclusion 2 is based on removing the GPS ICs from the GPS series, i.e., conclusions 1 and 2 are totally unrelated. The GPS ICs were obtained from GPS series that were previously detrended, explaining the small change of the estimated velocities from the filtered series. The ICA filtering also explains the reduction of the noise in the series and, therefore, of the estimated velocity uncertainty from the filtered series.**

The comment about removing the modeled loading from the GNSS time series was made also by Referee #1, we answer as follows:

Our goal is to remove signals associated with meteo-climatic processes using vbICA, instead of subtracting modeled displacements, such as those made available through loading services like GFZ, from the measured displacements. This approach minimizes biases due to the mismatch between the actual signal caused by atmospheric and hydrological loading and the modeled ones. Larochelle et al. (2018) reached similar conclusions by comparing GRACE measurements and the results from ICA decompositions of GNSS displacements, which resulted to be more accurate in correcting GNSS from seasonal displacements than removing GRACE displacements, which smooth local effects in the data acquisition and processing.

This is now described at the beginning of Section 5.1:

*"Our goal is to estimate the tectonic velocity of the GNSS stations, then we seek to remove signals associated with meteo-climatic processes. Instead of subtracting from the IGb14-time series the modeled displacements, such as those made available through loading services like GFZ, we prefer to subtract the displacements associated with the ICs. This approach minimizes biases due to the mismatch between the actual signal caused by atmospheric and hydrological loading and the modeled ones. Larochelle et al. (2018) reached similar conclusions by comparing GRACE measurements and the results from ICA decompositions of GNSS displacements, which resulted to be more accurate in correcting GNSS from seasonal*

*displacements than removing GRACE displacements, which smooth local effects in the data acquisition and processing.*

*In order to support the approach followed, we estimated the scatter of the GNSS displacement time series by computing the mean standard deviation of 1) the time series given as input to vbICA (IGb14-time series), 2) the IGb14-time series minus the combined displacement associated with the first 3 ICs and 3) the IGb14-time series minus the displacements due to HYDL+NTAL from GFZ models. The resulting standard deviation is 5.32, 4.10 and 4.73, respectively. This demonstrates that removing the displacement associated with the first four ICs is more effective in reducing the scatter than removing the HYDL+NTAL contribution."*

**Where I think this approach fails is that the raw series (used to estimate the velocity, the filtered velocity being very similar) and the filtered series (used to re-estimate the velocity uncertainty) are not consistent and therefore the velocity and its "improved" uncertainty are not consistent either. The authors could have tried a more aggressive filtering, like a band-pass filter leaving the trend and high-frequency noise only, or could have not consider colored noise in the velocity estimation (both ways are equivalent) and they will get even smaller velocity uncertainties. Unfortunately, this will not give any valuable information on the quality of the velocity and your ability to extrapolate it to understand tectonic physical processes. The only way to improve velocity estimates is to understand and reduce variability in the GPS series with proven corrections and models. If the white noise is more visible in the filtered series is probably because the GPS ICs absorb together a significant portion of the power-law noise that typically dominates the variance of the detrended GPS series, though this is not very clear from the IC PSDs in Fig. 3. Precisely, the power-law noise in the GPS series is only mentioned briefly and its influence on the GPS ICs and on the correlation with the loading ICs is not discussed at all.**

> We do not understand what the reviewer means by "not consistent". Filtering Common Mode Signals or Common Mode Errors from GPS time-series is a very commonly adopted step, performed in many different ways (and our work discusses one possible approach) when research topics require improvement of the signal to noise ratio, and there is a vast literature on that. We agree that filtering the time series by applying, for example, a pass-band filter, does not give any valuable information on the quality of the velocities, and, in particular, on the nature of the signals filtered out. We partially disagree with the sentence that "the only way to improve velocity estimates is to understand and reduce variability in the GPS time-series with proven corrections and models". We agree that the common goal is to reduce the "variability" in daily positions, but adopting "proven corrections and models" is one way to reach that goal, not the "only way". For example, Dong et al. (2002) well described how to reduce variability in GPS time-series by adopting several models, however, in more recent papers similar results have been obtained by applying multivariate statistical methods (eg., Tan, Chen, Dong et al., Remote Sensing, 2020; Yan, Dong et al. JGR, 2019; Tan, Dong and Chen, Advances in Space Science, 2022, to cite a few from the same author). Larochelle et al., 2018, also, found that an approach that uses results from multivariate statistical methods is more accurate in filtering out seasonal signals than using proven models. See also our response to the next comment.
>
> We discuss the effect of time series filtering on the noise in lines 432-435, 484-495 and in Figure 11.

**With respect to the GPS ICs and their attribution of a geophysical origin, I enumerate below several points raising concerns on the authors' approach. Generally, many past publications have shown than GPS series and loading models do not see the same thing, except partly for the annual variation.**

This latter sentence would provide further justification for the approach we have used in this manuscript. However, we agree that comparisons between models and geodetic observations greatly help in the interpretations, and this is exactly what we have done in Section 4.2.

**Most of the variance in the loading model series is concentrated at the annual period. Compared to the PSD of the loading models, the GPS series contain a relatively higher variance at long periods with a distinct PSD slope and a PSD much richer in periodic artifacts at short periods. The authors briefly comment on the systematic errors that are present in the GPS series, but they do not try to make the GPS series more consistent with the model series. For instance, it is known the annual draconitic variation could significantly affect the comparison to the solar annual variation of the loading models.**

The analysis of the PSD does not show the presence of frequencies associated with the draconitic signal in any of the ICs. While the draconitic signal frequency (~1.04 cycles per year) might be hidden in the annual frequencies of the ICs resulting from the GNSS data analysis (Fig. 3), it is much less relevant to explain data variance then the processes we discussed to interpret the GNSS_ICs (temperature-related processes, atmospheric and hydrological loading), whose frequencies are exactly 1 cycle per year. Otherwise, we would have observed an IC with a PSD of ~1.04 cycles per year, but this is not the case.

**The results obtained by the authors are confusing (see points below) and do not refute findings from past publications, contrary to their claims to successfully separate geophysical signals from the GPS series. For instance, authors show no evidence that the HYDL series significantly explain variations in their GPS series. The GPS and NTAL annual seem to partly agree (see points below), so the authors introduce a thermal annual component in the discussion without providing strong evidence nor explanation of its spatial pattern.**

NTAL is the most relevant cause of the displacements observed by GNSS, but also HYDL plays a role. This is proven by the correlation between HYDL and GNSS and by the increasing correlation with GNSS when considering NTAL+HYDL instead of HYDL only.

We are not introducing a thermal annual component, IC4 is a result of the vbICA analysis. We tried to better explain and interpret its spatial pattern in the main text:

*"Air temperature increase can induce both positive and negative vertical displacements. In the alpine valleys the water content increases as the temperature increases because of the snow and ice melting. It follows that in those areas the elastic hydrological load is higher during summertime than winter (Capodaglio et al., 2017), so that negative vertical displacements are observed when the temperature increases. Then, it is not surprising that in the alpine valleys the stations affected by large IC4-related displacements move downward as temperature increases. This may be an example of a small-scale hydrological process that is likely badly reproduced by the global HYDL, which does not have a spatial resolution fine enough to represent hydrological loading displacements at the scale of the alpine valleys. Other site-dependent processes that can potentially induce uplift during winter are the ice formation, and subsequent melting, in the antenna and antenna mount (Koulali and Clarke, 2020) and soil freezing (Beck et al., 2015).*

*Conversely, positive vertical displacements as the temperature increases can be caused by monument/bedrock thermal expansion and by the drying of the soil, because of the reduction of the elastic hydrological load. While HYDL takes into account the drying of the soil, we cannot exclude that some very local, unmodeled, environmental conditions can amplify this effect at some sites. This might explain why most of the sites affected by uplift during temperature*

*increases are located in plain areas, like the northern sector of the Paris Basin and in the Po plain, instead of the mountainous ones.*

*The relation between IC4 and local processes is also suggested by the heterogeneity of this signal in terms of spatial distribution, sign, amplitude and relevance in explaining the data variance. In fact, while ~50% of the station have U4<2mm [...]"*

**It is also probably worth mentioning that, if the GPS series were effectively explained by the combination of atmospheric/hydrology loading and temperature variations, as the authors claim, we should get the same GPS series out of the same GPS data when using different software, different strategies and different corrections. However, this is often not the case, especially when comparing global and regional GPS solutions.**

Comparing the GPS solutions obtained using different software and strategies is out of the scope of this work. Section 5.2 is updated with additional content that we think helps to interpret the correlations shown in Section 4.2 between GNSS_ICs and NTAL+HYDL_ICs. Furthermore, in the introduction we point out that

*"Excluding tectonic and volcanological processes, and once removed the effect of tides associated with solid earth, pole and ocean, variations of atmospheric pressure loading and fluid redistribution in the Earth crust are the main cause of vertical ground displacement recorded by GNSS stations worldwide (Liu et al. 2015)"* .

It follows that we are not surprised to find the contribution of HYDL+NTAL in our GNSS data.

**Other general points:**

**1) While I understand the objective of the ICA applied to the GPS series is to separate the variability into independent processes, I cannot understand the rationale for applying ICA to the NTAL and HYDL series. What are the independent processes to be separated in the atmospheric pressure loading or water loading? Even more confusing are the results from the comparison of a single GPS IC to a single NTAL/HYDL IC and the claim that the GPS series are explained by both.**

The goal of applying vbICA to HYDL and NTAL time-series is not that of separating possible different sources, but to investigate the presence of possible spatial and temporal signatures in the model datasets to be compared with results from GPS decomposition, such as the ones discusses in this work (IC1, IC2 and IC3).

At the beginning of 4.2 we add some text to better and, hopefully, more clearly explain why we decompose NTAL and HYDL and what the different component mean in terms of "sources":

*"As discussed in the introduction, atmospheric and hydrological loading are likely the main sources of vertical displacement in the great Alpine region. Since they are both uniform in terms of spatial response, showing smooth spatial variations, we decided to check if the first 3 ICs of the GNSS decomposition are associated with the displacements due to atmospheric and hydrological loading, and with their pattern of variability.*

*The vbICA analysis separates the data into statistically independent signals, which is useful because independent signals are often caused by different and independent sources of deformation. Nonetheless, a single source of deformation, such as atmospheric or hydrological loading, can be spatially heterogeneous and characterized by peculiar spatio-temporal patterns. In this case, the vbICA separates a single source of deformation in different signals associated with different spatio-temporal patterns. As a consequence, we decided to apply a vbICA decomposition on HYDL and NTAL model displacement time series in order to check if*

*they show any pattern and if they resemble the spatial distribution of IC1, IC2 and IC3 of the GNSS decomposition."*

**The ICA analysis is forcing the NTAL/HYDL series into non-gaussian independent components, even if they do not exist physically. This probably explains why the total NTAL annual is split across ICs with spatial patterns as orthogonal as possible.**

It is worth noting that the similarities between the spatial patterns of the NTAL and HYDL independent components do not depend on how the algorithm works: the relative position of the sites is never taken into account during the analysis.

The presence of N-S and E-W gradients in the ICs of both NTAL and HYDL, and also of the precipitation data, is likely caused by their link to a common, meteo-climatic, source. In fact, precipitation, atmospheric and hydrological loading depend on the climatic conditions, which are spatially and temporally variable. Besides IC1, which is a spatially uniform signal explaining more than the 90% of the total variance in either NTAL or HYDL decomposition, IC2 and IC3 probably reveal the spatio-temporal features of the weather regimes that cause atmospheric and hydrological loading on the surface: the Atlantic Ridge and the North Atlantic Oscillation. In section 5.2 we added the following part:

*"It is then likely that weather regimes like the NAO and the Atlantic Ridge influence both NTAL and HYDL, which is mainly forced by precipitation, so that the spatial patterns of the ICs associated with atmospheric and hydrological loading are the same of NAO (N-S) and Atlantic Ridge (E-W)."*

**The same spatial patterns are found for the GPS series, probably because once the trend, offsets and annual are removed from the GPS series, what is left is a Gaussian or near Gaussian series with temporal & spatially correlated noise and also the above-mentioned systematic periodic errors. It may be that the easiest way for the ICA to force the separation of these residual series into ICs is by making their spatial patterns orthogonal (see another possible explanation in point 5 below). The authors' conclusion that GPS and loading see the same spatial patterns is therefore not very solid.**

We would agree if the N-S and E-W patterns weren't found in NTAL and HYDL. Anyway, since we observe these kinds of patterns, and there is also temporal correlation between NTAL+HYDL_IC2(3) with GNSS_IC2(3) what we are observing is more likely a signal than noise. See also the updated Figure 7.

[Figure]

*Figure 7: Comparison, at the LYSH (Lon: 18.45°; Lat: 49.55°) site, between the displacements associated with: a) GNSS_IC1 and NTAL+HYDL_IC1; b) GNSS_IC2 and NTAL+HYDL_IC2 ; c) GNSS_IC3 and NTAL+HYDL_IC3. d), e), f) are the same as a), b), c), respectively, for the STV2 (Lon: 6.11°; Lat: 44.57°) site. A 30-days moving average filter is applied to better visualize the data.*

**2) The GPS and NTAL/HYDL series have different spatial samplings, which must complicate the interpretation of their comparison. Also related to the spatial sampling, it must be difficult to extract accurate NTAL values in the Alps due to the pressure model resolution and the short-scale changes in topographic gradient, making its comparison to the GPS series even less trustworthy. I suspect similar limitations exist when comparing GPS and HYDL model series in a mountain range.**

We do agree, and we are aware that at some GNSS sites probably HYDL do not correctly model the displacements caused by hydrological loading because of very local scale processes. We make that more explicit in the text when discussing the interpretation of IC4:

*"In the alpine valleys the water content increases as the temperature increases because of the snow and ice melting. It follows that in those areas the elastic hydrological load is higher during summertime than winter, as observed by Capodaglio et al. (2017), so that negative vertical displacements are measured when the temperature increases. Then, it is not surprising that in the alpine valleys the stations affected by large IC4-related displacements move downward as temperature increases. This may be an example of a small-scale hydrological process that is likely badly reproduced by the HYDL displacement dataset, which does not have a spatial resolution fine enough to represent hydrological loading displacements at the scale of the alpine valleys. Other site-dependent processes that can potentially induce uplift during*

*winter are the ice formation, and subsequent melting, in the antenna and antenna mount (Koulali and Clarke, 2020) and soil freezing (Beck et al., 2015)."*

On the other hand, the concordance between NTAL and GNSS time series seems very good, in particular when considering IC1 (Fig. S8), which explains the largest percentage of variance of the data. The overall agreement between HYDL+NTAL with the displacements associated with the first 3 ICs seems robust to us and we believe that this justify the approach of estimating environmental-induced displacements directly from the data and not from the models, which are used only for comparison.

**3) Each dataset used by the authors is decomposed in different numbers of ICs: 7 for GPS, although only 4 are discussed, and 3 for the model loadings. Then they compare the first 3 individual ICs and find weak correlations between them.**

IC5, IC6 and IC7 are discussed in the Supplementary Material, as we find that these are more localized features associated with local processes, not of interest for the Alpine area.

We do not agree with the word "weak" to define the correlation between the first 3 ICs of the GNSS and of HYDL+NTAL (Fig. 6). We provide additional details about the correlation between the displacements associated with the ICs, including not only the Lin, but also the Pearson correlation coefficient as suggested by Referee #1.

**The authors conclude on the origin of the individual GPS ICs based on their correlation to the individual loading ICs. However, this criterion is very weak, especially with correlation values around 0.6. As an example, similar (Pearson's) correlation values would be obtained between a pure sinusoidal and the same sinusoidal delayed almost pi/3, which is roughly two months if the sinusoidal has a period of one year. When subtracting one sinusoidal from the other, it is clear that we are not correcting much. The ratio of explained variance between the different ICs would have been more appealing, but, it is not clear that the individual ICs from different datasets correspond to the same fraction of the total signal (see point 1).**

We added the percentage of explained variance in the figures.

We show histograms with the maximum displacement associated with each IC in the supplementary material (Fig. S4, which we now moved in the main text as suggested by Rev#3), while the spatial response in Fig. 3 shows the displacement associated with each station for each IC.

We agree that two sinusoidal signals can be correlated even if they are out of phase. Nonetheless in our case, especially when considering the displacements associated with atmospheric loading, which are larger than the ones caused by atmospheric loading, we observe temporal evolutions (Fig. 4) which are far from a pure sinusoid. It follows that it is very unlikely that a signal is by chance correlated, both in terms of amplitude and temporal evolution, with the ones shown in Fig. 4. Further evidences about that are shown in Fig. 7, where the correlations between IC1 (Fig. 7a 7d), which are around 0.6, are not the result of two sinusoids out of phase.

**So maybe the ICA method is not well adapted to this problem or should not be applied to the NTAL/HYDL series (see point 1). A band-pass filtered comparison of GPS and loading series would probably be more informative here. Also rather than filtering the GPS series, I think it would have been better if the authors had shown how the loading models change the variance of the GPS series, as it is done in many other publications. The loading would need to be computed**

**at the station locations. It would have been even better to show how the GPS variance changes (not necessarily reducing) all along its power spectrum when correcting the loads.**

We estimated the scatter of the time series by computing the mean standard deviation of the time series given as input to vbICA (IGb14-time series); IGb14-time series minus the combined displacement associated with the first 3 ICs; IGb14-time series minus the displacements due to HYDL+NTAL. The resulting standard deviation is 5.32, 4.10 and 4.73, respectively. This demonstrates that removing the displacement associated with the first four ICs is more effective in reducing the scatter than removing the HYDL+NTAL contribution.

**4) The authors are processing a regional network and aligning it to a global linear frame (IGb14) that does not include seasonal variations. The frame alignment of the daily solutions from regional networks acts as another CME-like filtering of the series, not discussed by the authors, but probably similar to the SFM method. The filtering is more efficient as the network size is smaller, but the authors do not provide enough information on this point. It is then difficult to interpret the common network-wide annual signal shown by the GPS IC1. I would expect the regional frame alignment would absorb part of this common GPS annual signal, making it difficult to compare to the loading model and also leaving an amplitude much smaller than the residual station-dependent annual signal that is probably captured by the IC4. However, the numbers in table 1 indicate the opposite, assuming the average "of the amplitude of the maximum displacement" is somehow related to the annual amplitude, which is not clear either. The annual variation is the most prominent signal in NTAL with amplitudes typically of a few mm, less than 1 cm at the center of large continental masses. So it's not clear what the authors mean with atmospheric loading amplitudes larger than 2 cm. It is also not mentioned which frame was used to create the loading series and whether they were detrended like the GPS series, especially the HYDL series.**

The position time-series used in this work do not come from a regional GNSS solution. As explained in Section 3.1 and in the Supplementary Material, the Alpine time-series are part of a much larger solution that includes data from >4000 continuous GNSS stations distributed mainly in the Eurasian and African plates. It is worth considering that seasonal terms in ITRF have been introduced only with ITRF2000. The IGb14 frame is determined by a robust "quasi-global" network of ~250 IGb14 core sites + some regional high-quality stations (see figure below, where blue circles show the sites used to define the IGb14 reference frame). For this reason, we are quite confident that if some CME is absorbed by the daily alignment to IGb14, this is a fraction of the one in case of regional solutions. The N-S and E-W gradients in spatial patterns of common ground displacement components were found in Serpelloni et al. (2013), who used a continental-scale solution, by combining regional solutions with global MIT SINEX and using 246 IGS stations for the reference frame definition (Fig. S1). Figure S1 is now included in the Supplementary Material.

We added, from line 209, a more detailed explanation on how to interpret the temporal evolution, the spatial distribution and the displacement associated with the ICs.
*"Before discussing the vbICA results, we briefly explain how to interpret the temporal evolution and the spatial distribution of the ICs, so that it is possible to retrieve the displacements associated with them.*
*The color of each GNSS site in Fig. 2 represents the IC2 spatial response (U2), which indicates the maximum displacement associated with the IC2, while the temporal function V2 is normalized between 0 and 1. The displacement associated with IC2 between two epochs (e.g. t1*

*and t2, with t2>t1) at the station n is computed as V1(t2)\*U1n-V1(t1)\*U1n(t1), where V1(t2) is the value associated with the temporal evolution of the IC at the epoch t2.*

*U1n depends on the site, but not on the epoch; its unit of measurement is mm, while V has no units of measurement. As a result, V1\*U1n is in mm. It follows that if U1n is positive, as we observe for each station, and V1 is increasing (V1(t2)>V1(t1)), the stations move upward during the t2-t1 time interval. On the other hand, if V1(t2)<V1(t1) the stations move downward during t2-t1.*

*As regards Fig. 2, assuming t1=2010.0 and t2=2020.0, the displacements associated with IC2 are ~30 mm upward at the "red" GNSS stations, ~30 mm downward at the "blue" GNSS stations and ~0 mm at the white ones."*

We also modify lines 255-256:

*"IC1 is a spatially uniform signal characterized by an annual temporal signature, as shown by the power spectral density (PSD) plot in Fig. 3a. The mean of the maximum amplitudes is 26 mm, while the histogram showing the distribution of displacement amplitudes is shown in Fig. S3a.*

*IC2 shows a spatial response characterized by a clear E-W gradient, but, differently from IC1, its temporal evolution has not a dominating frequency. The spatial response U2 of the eastern stations (in blue) is mainly negative, while the U2 of the western stations (in red) is mainly positive."*

We used the Center of Figure reference frame and the time series were not detrended; we added this information in the text.

[Figure]

*Figure S1: Distribution of the continuous GNSS stations analyzed (red dots). The blu circles show the sites used to define the reference frame, including all IGb14 core sites, integrated by additional, high-quality, IGS stations.*

**5) The 2nd and 3rd GPS ICs are particularly interesting. These represent daily E/W and N/S network tilts with a rather flat spectrum. The NTAL and HYDL show similar spatial tilts, but their physical meaning is dubious (see point 1) and their spectral content is completely different: mostly seasonal for NTAL and mostly interannual for HYDL. The origin of these network tilts is very likely not the same among the datasets, as stated by the authors. In addition, if the whole GPS network is truly moving like these two ICs and it is not an artifact of the ICA separation, I would first think of a problem with the reference frame alignment. As said in point 4, network-wide common mode signals, including daily tilts and annual up & downs, should be at least partly (if not totally) absorbed by the frame alignment as these signals are not included in the linear reference frame and the network size is probably not large enough. Figure 7b must be wrong as there is no annual variation in the GPS IC2.**

We updated Fig. 7 considering different stations: one located in the south-western part of the network (STV2), the other in the north-eastern side (LYSH), so that the displacement associated with GNSS_IC2 and GNSS_IC3 have opposite sign.

In section 5.2 we discuss with more details the interconnection between precipitation, atmospheric pressure and hydrological loading. We hope that this part helps to make more clear that what GPS is recording is mostly caused by the environmental contribution (atmospheric + hydrological loading) and not by data processing errors like the reference frame alignment or the draconitic variation:

"*We also performed a vbICA analysis on precipitation data (RAIN) recorded over the study region, using 3 ICs. The spatial pattern of the ICs is analogous to the ones associated with NTAL and HYDL (Fig. 14).*

[Figure]

*Figure 14: IC1, IC2 and IC3 of the RAIN decomposition.*

*This supports the hypothesis that precipitation, atmospheric pressure, hydrological loading and ground displacement are somehow interconnected and characterized by a common climate-related forcing, whose characteristics of spatial variability are described by the NAO and Atlantic Ridge weather regimes.*

We point out that HYDL, NTAL and GNSS are models or measurements of vertical displacement, which is positive when upward and negative when downward; while RAIN is the amount of fallen rain per unit area.

Let us consider for the sake of simplicity the IC1 case, but what we are going to discuss holds true also for IC2 and IC3.

The temporal evolution of NTAL_IC1 (NTAL_V1) is correlated with the temporal evolution of RAIN_IC1 (RAIN_V1, Fig. 15g-i) and anti-correlated with the time derivative of the temporal evolution of HYDL_IC1 (HYDL_V1, Fig. 15a-c). HYDL_V1 is also highly anti-correlated with RAIN_IC1 (Fig. 15d-f).

[Figure]

**Figure 15: Cross correlation between:**
**a) the temporal evolution of the IC1 of the NTAL decomposition and the time derivative of the temporal evolution of the IC1 obtained by decomposing HYDL; b) same as a), but considering IC2; c) same as a), but considering IC3;**
**d) the temporal evolution of the IC1 of the precipitation data decomposition and the time derivative of the temporal evolution of the IC1 obtained by decomposing HYDL;  e) same as d), but considering IC2; f) same as d), but considering IC3;**
**g) the temporal evolution of the IC1 of the NTAL decomposition and the temporal evolution of the IC1 of the precipitation data decomposition; h) same as g), but considering IC2; i) same as g), but considering IC3.**

Our interpretation of the correlations discussed above, schematically represented in Fig. 16 is the following: when the weather goes from a low pressure to a high pressure regime, the increasing pressure causes a downward displacement of the ground (Fig. S8). Anyway, low pressure regimes are often associated with precipitation, and that is why IC1_RAIN and IC1_NTAL are correlated. It follows that when we go from high pressure to low pressure conditions, the ground motion, if we assume a pure elastic process, is affected by two forces acting in opposite directions: the decreasing atmospheric pressure induces uplift, while the precipitation load causes downward motion. Rain also affects hydrological loading, increasing it and causing a downward ground motion. As a consequence, the temporal derivative of HYDL_IC1, which is more sensitive to small but fast variation of hydrological loading than HYDL itself, is negative and anti-correlated with IC1_RAIN.

[Figure]

***Figure 16:** Schematic representation of the ground vertical displacement due to elastic deformation during high pressure (a) and low pressure (b) conditions. Yellow arrows reflect displacements associated with atmospheric pressure, blue arrows reflect displacements associated with precipitation and evapotranspiration.*

*Atmospheric pressure variations happen at fast temporal scales, then the switch from high to low pressure conditions (and vice versa) can happen in a few days and cause quite large (centimetric) ground vertical displacements. Hydrological loading acts at longer timescales and there are several factors to consider besides precipitation, in particular the temperature, which causes evapotranspiration. Nonetheless, computing the time derivative of the hydrological loading allows to detect "fast" variations due to the change of the atmospheric pressure and the precipitation events often associated with it."*

---

## Author Comment (AC3)

The manuscript "Common mode signals and vertical velocities in the great Alpine area from GNSS data" by Francesco Pintori et al. presents how ICA decomposition of GNSS time series in the alpine area allows to separate sources of deformation and then retrieve with a better uncertainty the velocity field in Europe. The authors process the daily GPS observations with GAMIT/GLOBK software, using subnetworks later tied to IGb14 reference frame. The obtained 2010-2020 time series have then been analysed in order to explore the origin of the common modes, and the potential of Independant Component analysis to extract these modes with a more "physical" basis and filter the time series. The ICA method used in the paper is the vbICA, a bayesian multivariate source separation method. The ICA analysis conducted here is performed in two steps, one, with 8 components, allows to extract and correct the trend (the velocity), the other, with detrended GNSS data as input, contains seven components. In parallel, hydrological and atmospheric loading predictions from two institutes are also analysed with vbICA with three components. These three components corresponds mostly to a uniform spatial pattern, an E-W trend and a N-S trend. The GNSS components appear well correlated to the hydrological plus atmospheric loads components, proving the loading origin of these components. A last component is clearly seasonal and presents spatial variation at small wavelength, in phase with temperature variations. The four vbICA components are used to correct the GNSS time series, which allow a new estimation of the velocity, in very good agreement with the first estimation but with a much smaller error estimation. The authors also compare different methods for common mode estimation, the stacking Filtering method, or weighted stacking filtering method to the filtering obtained by an Independant component analysis.

Overall I found the manuscript interesting and worth of publication, as it shows a convincing correspondance between what is referred as "common modes" and the atmospheric and hydrological loading. However, I think that the paper, although well written, is quite hard to follow, with numerous abbreviations, and comparisons which could be better presented and illustrated. I have also a few scientific comments that can be adressed. I suggest a major revision.

Here are my suggestions:

* I find intriguing that the main three components that are discussed here correspond to a uniform pattern, an E-W tilt and a N-S tilt. These three components correspond to the largest perpendicular spatially correlated signals possible.

(1) Can you change the color scale of all panels of IC1, to show how uniform it really is ? For example GNSS IC1 should be plotted with a 20-32 scale.

    Ok, we changed Figures 3, 4, 5.

(2) For IC2 and IC3, how significantly different from a tilt the components are?

    To answer this question we performed a Principal Component Analysis (PCA) on IC2, IC3 and IC4 data: we generated a 545x3 matrix of data where each row is associated with a GNSS stations and the three columns are the corresponding longitude, latitude and spatial response (U). Longitude and latitude have been converted into km to avoid distortions and U has been multiplied by a weighting factor, so that its amplitude has the same order of magnitude of the

longitude. The PCA on those data allows us to estimate how well two PCs, which define a tilted plane (Fig. R1), represent U2, U3 and U4.

The variance explained by the plane associated with the first two PCs is:
  - 97.7% for U2;
  - 97.0% for U3;
  - 83.8% for U4.

This shows that U2 and U3 are both well approximated by a tilt; in fact, the percentage of explained variance is very similar and larger than IC4, which does not have any tilt features.

[Figure]

*Figure R1: Representation of the tilted planes defined by PC1 and PC2 used to fit U2 (a); U3 (b); U4 (c).*

It is worth noting that this result does not depend on how the algorithm works: the relative position of the sites is never taken into account during the analysis.

The presence of N-S and E-W gradients in the ICs of GNSS, NTAL, HYDL and precipitation data is caused by their link to a common, meteo-climatic, source. In fact, precipitation, atmospheric and hydrological loading depend on the climatic conditions, which are spatially and temporally variable. Besides IC1, which is a spatially uniform signal explaining more than the 90% of the total variance in either NTAL or HYDL decomposition, IC2 and IC3 probably reveal the spatio-temporal features of the weather regimes that cause atmospheric and hydrological loading on the surface: the Atlantic Ridge and the North Atlantic Oscillation. In section 5.2 we added the following part:

*"It is then likely that weather regimes like the NAO and the Atlantic Ridge influence both NTAL and HYDL, which is mainly forced by precipitation, so that the spatial patterns of the ICs associated with atmospheric and hydrological loading are the same of NAO (N-S) and Atlantic Ridge (E-W)."*

**(3) the loading models appear to predict mainly very long wavelength features, corresponding to the first three components. Is this true?**

Yes, it is true. Since the loading models are global, evaluated over a grid with a spatial resolution of 0.5°, they do not have a great spatial resolution. It follows that it is easier to observe long wavelength features instead of the local ones.

**Can you show an example of the predicted load-induced displacement map?**

Since the displacements associated with both HYDL+NTAL are not the same over the study area, we cannot show a load-induced displacement map. Nonetheless, we can compute the displacement due to HYDL+NTAL models in some specific GNSS sites. For example, in Figure 7 we compare, at two GNSS sites, the displacements associated with the GNSS_ICs and with the HYDL+NTAL_ICs.

[Figure]

*Figure 7: Comparison, at the LYSH (Lon: 18.45°; Lat: 49.55°) site, between the displacements associated with: a) GNSS_IC1 and NTAL+HYDL_IC1; b) GNSS_IC2 and NTAL+HYDL_IC2; c) GNSS_IC3 and NTAL+HYDL_IC3. d), e), f) are the same as a), b), c), respectively, for the STV2 (Lon: 6.11°; Lat: 44.57°) site. A 30-days moving average filter is applied to better visualize the data.*

Furthermore, in Figure R2 we show the results of the ICA decomposition of the displacements associated with the combined contribution of atmospheric and hydrological loading (HYDL+NTAL), as you also suggest in a comment below, and in Figure R3 the Lin correlation coefficients between: a) GNSS-IC1 and NTAL+HYDL_IC1; b) GNSS_IC2 and NTAL+HYDL_IC2; c) GNSS-IC3 and NTAL+HYDL_IC3.

[Figure]

***Figure R2:** ICA decomposition, using 3 components, of the displacements associated with the combined contribution of atmospheric and hydrological loading (HYDL+NTAL).*

[Figure]

*Figure R3: Using the results of the ICA decomposition on the displacements associated with the combined contribution of atmospheric and hydrological loading (HYDL+NTAL) represented in the figure above (Fig. R2), we show the Lin correlation coefficients between: a) GNSS-IC1 and NTAL+HYDL_IC1; b) GNSS_IC2 and NTAL+HYDL_IC2; c) GNSS-IC3 and NTAL+HYDL_IC3. Histograms of the correlation coefficients are also reported.*

**The percentage of the variance do the three components is indicated to be > 97%. For atmosphere, I guess pressure variations are large-scale such that the earth response is also at large-scale. But I would have thought that hydrological loading should be more local. Can you comment on that ?**

We do agree that hydrological loading is more sensitive to local processes than atmospheric pressure. Nonetheless, we use the results of the global models to estimate the hydrological loading, even though we are aware that some local effects might not be captured. In fact, considering the extension of the study area, it is very complicated to take into account the local

features needed to estimate the hydrological loading with a better precision than the one provided by the global models.

**\* The seasonal contribution should not be named temperature contribution. This would suggest a thermal contraction effect which is far from being proven. A lot of signals could be seasonal. Unless you prove that there is a strong correlation between the IC4 and temperature beyond the seasonal term (ie at higher frequency) the correlation appears fortuitous. Fig 8 shows that temperature seems to have higher frequency fluctuations not observed in IC4, but it s hard to tell from the figure only.**
**I suggest to rewrite the paragraphs and sentences related to this seasonal contribution of unknown origin everywhere in text.**

We agree that the conclusions on IC4 are too strong. We have modified the abstract, changing a sentence that incorrectly let the reader suppose that temperature might directly cause the displacement associated with IC4.

In fact, it is more correct to state that the displacements could be caused by processes correlated to temperature, which are discussed in section 5.2, than caused by temperature itself.

The lines 504-516 are also updated, discussing some hypothesis about the correlation between temperature and the displacement signal associated with IC4:

*"Air temperature increase can induce both positive and negative vertical displacements. In the alpine valleys the water content increases as the temperature increases because of the snow and ice melting. It follows that in those areas the elastic response to hydrological load is higher during summertime than winter, as observed by Capodaglio et al., (2017), so that negative vertical displacements are measured when the temperature increases. Then, it is not surprising that in the alpine valleys the stations affected by large IC4-related displacements move downward as temperature increases. This may be an example of a small-scale hydrological process that is likely badly reproduced by the HYDL displacement dataset, which does not have a spatial resolution fine enough to represent hydrological loading displacements at the scale of the alpine valleys. Other site-dependent processes that can potentially induce uplift during winter are the ice formation, and subsequent melting, in the antenna and antenna mount (Koulali and Clarke, 2020) and soil freezing (Beck et al., 2015).*

*Conversely, positive vertical displacements as the temperature increases can be caused by monument/bedrock thermal expansion and the drying of the soil, because of the reduction of the hydrological load. While HYDL takes into account the drying of the soil, we cannot exclude that some local, unmodeled, environmental conditions can amplify this effect at some sites. This might explain why most of the sites affected by uplift during temperature increases are located in plain areas, like the northern sector of the Paris Basin and in the Po plain, instead of the mountainous ones.*

*The relation between IC4 and local processes is also suggested by the heterogeneity of this signal in terms of its spatial distribution, sign, amplitude and relevance in explaining the data variance. In fact, while ~50% of the stations have U4<2mm (Fig. S3d) and explain <1% of the data variance, meaning that IC4 is almost unuseful to reproduce the original data, there is a non-negligible number of stations (~10%) explaining >10% of the data variance and with U4>6mm."*

**\* The statistics shown (mean, median, standard deviation) in tables and discussed in text are not well presented. I suggest to move S4 in the main text, it is quite graphical and shows better the**

**agreement in terms of distribution than Tables 1 and 2, that could be moved to supplementary material.**

Thanks for this suggestion, we move Tables 1 and 2 in the Supplementary and Figure S4 in the main text.

**lines 289 to 292 could be replaced by a more readable text.**

We changed lines 289-292, simplifying the text:
"*IC2 and IC3 of both NTAL and HYDL show E-W and N-S gradients in the spatial response, respectively, as observed for IC2 and IC3 of the GNSS dataset (Fig. 3b, d). Since the ICs spatial response of the NTAL and HYDL decomposition are very similar, we also consider the sum of the displacement associated with NTAL and HYDL models, which can be considered as "environmental loading": we use the notation NTAL+HYDL_ICn to indicate the sum of the displacement associated with the n-th component of the NTAL and HYDL decomposition. The amplitude of NTAL+HYDL_IC1, NTAL+HYDL_IC2 and NTAL+HYDL_IC3 are only slightly lower than the ones of GNSS_IC1 , GNSS_IC2 and GNSS_IC3, as shown in Fig. S4 (panels g,h,i) and in Table 1a.*"

**\* The part on correlation coefficients is confusing where it should not.**

We tried to make this part more clear

**If you consider that your signal is a sum of IC like $X_i(x,y)*T_i(t)$, then we expect to provide the correlation coefficient between Ti-GNSS and Ti-HYDR for example, or Ti-GNSS and Ti-ATM, and of Xi-GNSS with Xi-HYDR or XI-ATM. Only two values describing the temporal and spatial correlations would be sufficient. Here, it took me time to understand that, because you add $X_i\_ATM(x,y)*T_i\_ATM(t)$ and $X_i\_HYDR(x,y)*T_i\_HYDR(t)$, your spatial and temporal correlations stop being independant from each other. This is why I guess you provide ion Fig6 a spatial map of the temporal correlation of the GNSS and HYDR+ATM. Could you please clarify for the reader why you end up with such a plot ?**

If we consider the spatial and temporal correlations separately, we could miss some of the information contained in the data. The station by station computation of the Lin correlation between $X_i\_ATM(x,y)*T_i\_ATM(t)$ (or $X_i\_HYDR(x,y)*T_i\_HYDR(t)$) and $X_i\_GNSS(x,y)*T_i\_GNSS(t)$ allows us to take into account the amplitude of the displacement associated with each station. We would miss this information if we compared only the temporal evolution of the signals, as $T_i\_ATM(t)$ (or $T_i\_HYDR(t)$) with $T_i\_GNSS(t)$, by computing the Pearson correlation.
In Fig. 6 we add $X_i\_ATM(x,y)*T_i\_ATM(t)$ and $X_i\_HYDR(x,y)*T_i\_HYDR(t)$ and compare it, using the Lin correlation coefficient, with $X_i\_GNSS(x,y)*T_i\_GNSS(t)$. This allows us to associate the first three ICs of the GNSS decomposition, which have CMS features, with the displacement associated with the combined effect of hydrological and atmospheric loading.

**In fact, if you had made and ICA on (ATM+HYDR) directly, may be you would have obtained a similar result but easier to compare (ie an independent comparison in space and time).**

Thank you for this suggestion. The results (Fig. R3) are quite similar to what is shown in Fig. 6 and represent a good validation of what is shown in the main text.

We prefer not to add this in the main text because we decided to compute the HYDL+NTAL contribution only when we found that the ICs resulting from their decomposition have the same spatial patterns of the ICs associated with the GNSS data. We think that explaining why we decide to compute a-priori HYDL+NTAL could be harder to follow than what is written in the manuscript right now.

**The "blue points" on fig. 6 in the middle of the tilt, in opposite phase, have no real significance, as the spatial patterns of ICs do not exactly correspond to each other. I find more significant the peak in the ditribution, of 0.65 for IC2 and of 0.55 for IC3 which are significant numbers although the PSDs of the Ti do not really match.**

We do agree, in fact in Section 4.2 we point out that if we consider only the stations with amplitude associated with IC2 and IC3 larger than 3mm, the mean Lin correlation increases to 0.57 and 0.44, respectively.

**\* Once ATM and HYDR loads are proven to be good estimators of the common modes, why not use them to correct the time series ?**

Our goal is to remove signals associated with meteo-climatic processes using vbICA, instead of subtracting modeled displacements, such as those made available through loading services like GFZ, from the measured displacements. This approach minimizes biases due to the mismatch between the actual signal caused by atmospheric and hydrological loading and the modeled ones. Larochelle et al. (2018) reached similar conclusions by comparing GRACE measurements and the results from ICA decompositions of GNSS displacements, which resulted to be more accurate in correcting GNSS from seasonal displacements than removing GRACE displacements, which smooth local effects in the data acquisition and processing.

This is now described at the beginning of Section 5.1:

*"Our goal is to estimate the tectonic velocity of the GNSS stations, then we seek to remove signals associated with meteo-climatic processes. Instead of subtracting from the IGb14-time series the modeled displacements, such as those made available through loading services like GFZ, we prefer to subtract the displacements associated with the ICs. This approach minimizes biases due to the mismatch between the actual signal caused by atmospheric and hydrological loading and the modeled ones. Larochelle et al. (2018) reached similar conclusions by comparing GRACE measurements and the results from ICA decompositions of GNSS displacements, which resulted to be more accurate in correcting GNSS from seasonal displacements than removing GRACE displacements, which smooth local effects in the data acquisition and processing.*

*In order to support the approach followed, we estimated the scatter of the GNSS displacement time series by computing the mean standard deviation of 1) the time series given as input to vbICA (IGb14-time series), 2) the IGb14-time series minus the combined displacement associated with the first 3 ICs and 3) the IGb14-time series minus the displacements due to HYDL+NTAL from GFZ models. The resulting standard deviation is 5.32, 4.10 and 4.73, respectively. This demonstrates that removing the displacement associated with the first four ICs is more effective in reducing the scatter than removing the HYDL+NTAL contribution."*

**The advantage is that you can then anticipate that possible decadal trends of ATM and HYDR would then be removed from the time series and thus provide a better displacement rate due to tectonics. Here, the trend is first estimated from a first ICA, removed from GNSS time series, and then a new ICA is performed to extract ICs, that will correct the raw GNSS data, before a new trend estimation. How can you be sure that the last estimation will not be "by construction" biased towards the first ? On the other hand line 219-220 of 3.1 suggests that the separation of tectonics trend from other potential non tectonic trends is already done by the first ICA. Can you clarify this point ?**

As now reported in the conclusions, the procedure used in this work to estimate the station velocities does not allow to distinguish the tectonic velocities from the contribution to the velocity induced by climate-related processes, in particular if the linear trend associated with ATML and/or HYDL time series is large. Nonetheless, the small trend associated with HYDL_IC1 is likely the result of an annual signal whose amplitude is not constant over the years, which is captured by GNSS_IC1.

**Figures :**

**ICA figures:**

**- change color scales of IC1 for all plots to show lateral variations**

Ok, done.

**- temporal vector: normalisation should be made by variance and not by min/max (if I understood correctly) for the reader to visualie the relative amplitude of each term. Min/max can be outliers.**

We added, from line 209, a more detailed explanation on how to interpret the temporal evolution, the spatial distribution and the displacement associated with the ICs.
*"Before discussing the vbICA results, we briefly explain how to interpret the temporal evolution and the spatial distribution of the ICs, so that it is possible to retrieve the displacements associated with them.*
*The color of each GNSS site in Fig. 2 represents the IC2 spatial response (U2), which indicates the maximum displacement associated with the IC2, while the temporal function V2 is normalized between 0 and 1. The displacement associated with IC2 between two epochs (e.g. $t_1$ and $t_2$, with $t_2 > t_1$) at the station n is computed as $V1(t_2)*U1_n - V1(t_1)*U1_n(t_1)$, where $V1(t_2)$ is the value associated with the temporal evolution of the IC at the epoch $t_2$.*
*$U1_n$ depends on the site, but not on the epoch; its unit of measurement is mm, while V has no units of measurement. As a result, $V1*U1_n$ is in mm. It follows that if $U1_n$ is positive, as we observe for each station, and V1 is increasing ($V1(t_2) > V1(t_1)$), the stations move upward during the $t_2 - t_1$ time interval. On the other hand, if $V1(t_2) < V1(t_1)$ the stations move downward during $t_2 - t_1$.*
*As regards Fig. 2, assuming $t_1 = 2010.0$ and $t_2 = 2020.0$, the displacements associated with IC2 are ~30 mm upward at the "red" GNSS stations, ~30 mm downward at the "blue" GNSS stations and ~0 mm at the white ones."*

We also modify lines 255-256:

*"IC1 is a spatially uniform signal characterized by an annual temporal signature, as shown by the power spectral density (PSD) plot in Fig. 3a. The mean of the maximum amplitudes is 26 mm, while the histogram showing the distribution of displacement amplitudes is shown in Fig. S3a.*
*IC2 shows a spatial response characterized by a clear E-W gradient, but, differently from IC1, its temporal evolution has not a dominating frequency. The spatial response U2 of the eastern stations (in blue) is mainly negative, while the U2 of the western stations (in red) is mainly positive."*

Furthermore, Figures 3, 4, 5 are not characterized large outliers and we think that the min/max normalization is the most intuitive to show the displacements associated with the ICs.

**Figure 6: change colorscale to see changes in correlation coefficient for IC1 (the colorscale is completely saturated in the red).**

Ok, done

**Don't use "Lin" abbreviation but linear**

With "Lin correlation" we mean the Lin concordance correlation coefficient (Lin, 1989).

**Figure 7: panel b is identical to panel a**

Yes, we made an error.
We decide to change Figure 7, showing the displacement of two different sites, one located in the south-western part of the study region  (STV2), the other in the north-eastern side (LYSH), so that the displacement associated with GNSS_IC2 and GNSS_IC3 have opposite sign.

**Abstract**

**First sentence : too complicated. Simplify and clarify**

Ok, done.
"We study the time series of vertical ground displacements from continuous GNSS stations located in the European Alps. Our goal is to improve the accuracy and precision of vertical ground velocities, investigating the spatial and temporal features of the displacements caused by non-tectonic geophysical processes".

**line 10: associated with : modeled from**

Ok, done.

**line 11: processes: drop**

Ok, done.

**line 16-17 : Atmospheric .... gradients: rewrite**

Ok, done. Please note that we also modified lines 11-12:
"Furthermore, while the displacements caused by atmospheric and hydrological loading are apparently spatially uniform, our statistical analysis shows the presence of NS and EW displacement gradients."

**Introduction**

**First sentence: "active geophysical processes on land, ice and atomosphere": ground displacement on atmosphere. Rewrite.**

Ok, done.

**In general : a lot of references are missing on mountain uplift, both observations and mechanisms. Please provide some refs outside Italy.**

**Id. for lines 68-80**

Ok, we added the following references:
- Ching, K.-E., Hsieh, M.-L., Johnson, K. M., Chen, K.-H., Rau, R.-J., and Yang, M.: Modern vertical deformation rates and mountain building in Taiwan from precise leveling and continuous GPS observations, 2000–2008. Journal of Geophysical Research, 116, B08406, https://doi.org/10.1029/2011JB008242, 2011.
- Dal Zilio, L., Hetényi, G., Hubbard, J. and Bollinger, L: Building the Himalaya from tectonic to earthquake scales. Nature Reviews Earth & Environment, 2, 251–268, https://doi.org/10.1038/s43017-021-00143-1, 2021.

**line 117: give principle of CMC Imaging**

Ok, done. We added the following part:
*"A filtering method similar to CWSF, called CMC Imaging, is developed and used by Kreemer and Blewitt (2021) in western Europe to extract common mode components that are as local as possible. The main difference between CWSF and CMC Imaging is that the former uses as a weighting factor both the distance and the correlation coefficient among the stations, while the latter only the correlation coefficient, showing that it is representative of the distance among the stations."*

**line 190: pdfs --> PDFs (and elsewhere)**

Ok, done.

**line 192: drop "that"**

Ok, done.

**line 216: a priori any temporal : rewrite**

Ok, we rewrote the sentence as follows:
*"The advantage of this approach, compared to a trajectory model, is that it is not necessary to assume any temporal evolution of the deformation signals a priori, except for the limited number of functions that make up Eq. (1)"*

**line 389: k=-2 for both noise and flicker : correct text**

Ok, done.

**line 391: avoid + in text**

Ok, we use "plus" instead of +.

**line 506: elastic hydrological load ---> elastic response to hydrological load**

Line 506 is deleted in the updated version, but there we use "elastic response to hydrological load" in the text we added.

**\* Don't use "lin" abbrevation but replace by linear correlation coefficient.**

With "Lin correlation" we mean the Lin concordance correlation coefficient (Lin, 1989).

---

## Referee Report (RR1)

Pintori et al. 2021

The manuscript "Common mode signals and vertical velocities in the great Alpine area from GNSS data" by Pintori et al. 2021 utilizes variational Bayesian Independent Component Analysis to separate loading signals due to atmospheric and hydrologic sources from vertical GNSS time series to further isolate uplift rates due to mountain building in the greater alpine region. This also serves to provide additional information on the source of the common mode signals in GNSS time series. In this study, they also compare the vbICA method to other methods of constructing the common mode. Finally, they remove the non-mountain building signals from the data and estimate uplift rates. Overall, the article is well written and is a useful contribution and the revisions provided some clarification and improved the manuscript. However, there are a few aspects that would be improved with clarification and a few points the author overstate.

1.  In lines 188-189, you mention that the Bayesian approach introduces an approximating PDF for the parameters. Is the PDF chosen by the algorithm or by the authors? If the PDF is chosen by the authors as an input to the vbICA, does changing that PDF significantly change the output?
2.  For the NTAL and HDYL models, why are you modeling them at a grid interval rather than at the specific points of the GNSS stations? When you compare the ICs from the models to the GNSS stations (eg Figure 7), are you using the closest grid point or interpolating them to the GNSS station position? Would it not more consistent to just use the NTAL and HYDL models at the GNSS points and then apply the vbICA?
3.  This may be out of the scope of this paper, but I'm still wondering why the authors wouldn't remove known sources of loading, like NTAL, NTOL and HYDL and then apply a vbICA to see if there are any consistent patterns in the data not attributed to those well known signals? This could then highlight additional signals that cannot easily be removed from the data or are missed by the models (eg reference frame jitters). To me, this seems like a preferred method to use the vbICA since it would remove signals we know exist first and then diagnose the resulting signals. Otherwise, your ICs are likely to contain the known signals (NTAL and HYDL) as well as contributions from other signals that have similar temporal or spatial patterns – which the authors acknowledge when explaining why they combine multiple IC from different sources. Thus, the ICs are likely dominated by the loads you are attributing them to but also likely contain other signals that might be more apparent if you removed the known signal first.
4.  I still think the authors are overstating the temperature relationship in IC4. In lines 394, you state that it is well correlated with temperature, but couldn't you find a similar correlation with another annual signal, like NTAL for example or even HDYL? I think the added discussion in line 600 is helpful but still do not provide a strong enough case to be mentioned in the abstract and in the conclusion. Yes, IC4 has a strong annual signal but that does not necessarily mean that it is due to temperature fluctuations especially given that majority of the stations this component explains less then 1% of the data variance as stated in lines 618. Also, the mechanisms provided for temperature would be highly site specific and would be dependent on the type of monumentation (eg is the

monument located in bedrock vs unconsolidated soil monuments)? Additionally, the mechanisms provided are more related to hydrology and site characteristics then temperature. I think the relationship of IC4 to temperature is overstated especially since it is mentioned in the abstract and conclusion and multiple times throughout the text even though the IC is only prevalent in a few stations and the mechanisms provided do not support the claim.

5. Additionally, when calculating the reduction in the standard deviation, you only use the first 3 ICs (lines 477) and not IC4. Why?
6. You state the ICs likely contain a larger component of power-law noise. When you are comparing the different filtering approaches (line 477-479) when you remove the modelled GFZ NTAL and HYDL are you also estimating a noise model for those series as well?
7. Lines 433-435 are a touch confusing. Are you assuming that removing the ICs completely removes all the annual signal? Could there not be other processes that have annual signals that are not captured by the ICs?
8. Figure 6 might be improved by consistent axes.
9. Maybe I missed it, but what precipitation data are you using (lines 557).

---

## Author Response (AR2)

**1. In lines 188-189, you mention that the Bayesian approach introduces an approximating PDF for the parameters. Is the PDF chosen by the algorithm or by the authors? If the PDF is chosen by the authors as an input to the vbICA, does changing that PDF significantly change the output?**

The PDF of each variable is a priori chosen. Note though that to approximate the PDF of the sources (i.e., of the independent components) we are using a Mix of Gaussians (MoG). Given a sufficient number of Gaussians, it is possible to approximate any desired PDF. We restrain the number of Gaussians to 4 because in past research this number of Gaussians was sufficient to properly model the data retrieving accurate results (for the accuracy of the results, tests were performed on synthetic simulations for which the ground truth was known, see for example Gualandi et al., 2016, J. of Geodesy, and references therein). The final posterior PDF will thus be a MoG, and it can approximate multimodal distributions. What we can control, apart from the number of Gaussians in the mix, is the a priori on the Gaussians which govern their mean and precision (i.e., inverse of the variance). The results obtained using different a priori are overall similar, with differences in the variance explained of less than 0.07%.

Among the tested a-priori parameters we choose those that maximize the Negative Free Energy. This is equivalent to maximizing the independence of the sources according to the variational approach adopted that minimizes the Kullback-Leibler (KL)-divergence between the true (unknown) posterior and the modeled one.

**2. For the NTAL and HDYL models, why are you modeling them at a grid interval rather than at the specific points of the GNSS stations?**

GNSS data do not evenly cover the study region, while NTAL and HYDL models do. By using all the NTAL and HYDL data, it is easier to show the spatial patterns of the atmospheric and hydrological loading.

**When you compare the ICs from the models to the GNSS stations (eg Figure 7), are you using the closest grid point or interpolating them to the GNSS station position?**

When we compare the ICs from the models to the GNSS station we use the closest grid point, as we say in lines 349-351.

**Would it not more consistent to just use the NTAL and HYDL models at the GNSS points and then apply the vbICA?**

We agree that if we had used NTAL and HYDL models to remove the sources of loading from the GNSS time series, we could have considered only the data at the GNSS stations. Nonetheless, for the purpose of interpreting the spatial features of the GNSS results, it is preferable to use all the available data from the loading models, so that the spatial cover of the study area is as dense as possible.

**3. This may be out of the scope of this paper, but I'm still wondering why the authors wouldn't remove known sources of loading, like NTAL, NTOL and HYDL and then apply a vbICA to see if there are any consistent patterns in the data not attributed to those well known signals? This could then highlight additional signals that cannot easily be removed from the data or are missed by the models (eg reference frame jitters). To me, this seems like a preferred method to use the vbICA since it would remove signals we know exist first and then diagnose the resulting signals. Otherwise, your ICs are likely to contain the known signals (NTAL and HYDL) as well as contributions from other signals that have similar temporal or spatial patterns – which the authors acknowledge when explaining why they combine multiple IC from**

**different sources. Thus, the ICs are likely dominated by the loads you are attributing them to but also likely contain other signals that might be more apparent if you removed the known signal first.**

While we know that GNSS data are affected by hydrological and atmospheric loading, precisely quantifying their contribution is not an easy task. A mismodeling of the loading signal can lead to residuals in GNSS time series that may be very difficult to interpret; the residuals might also be temporal and/or spatially correlated, making even harder to see if there are any consistent patterns in the data not attributed to loading signals. We then prefer not to correct the GNSS time series with any model, that by definition cannot be perfect, but let vbICA extract the signals associated with the loading.

**4. I still think the authors are overstating the temperature relationship in IC4. In lines 394, you state that it is well correlated with temperature, but couldn't you find a similar correlation with another annual signal, like NTAL for example or even HDYL? I think the added discussion in line 600 is helpful but still do not provide a strong enough case to be mentioned in the abstract and in the conclusion. Yes, IC4 has a strong annual signal but that does not necessarily mean that it is due to temperature fluctuations especially given that majority of the stations this component explains less then 1% of the data variance as stated in lines 618. Also, the mechanisms provided for temperature would be highly site specific and would be dependent on the type of monumentation (eg is the monument located in bedrock vs unconsolidated soil monuments)? Additionally, the mechanisms provided are more related to hydrology and site characteristics then temperature. I think the relationship of IC4 to temperature is overstated especially since it is mentioned in the abstract and conclusion and multiple times throughout the text even though the IC is only prevalent in a few stations and the mechanisms provided do not support the claim.**

Following your advice we have eliminated the mention of the temperature in the abstract and in the summary. In line 618-619 we now say that IC4 might also be influenced by systematic errors in GNSS observations and in their modeling. In the conclusion, and in general over the manuscript, we present IC4 as a seasonal signal which is in phase with the temperature, but that is not necessarily caused by temperature itself.

**5. Additionally, when calculating the reduction in the standard deviation, you only use the first 3 ICs (lines 477) and not IC4. Why?**

In our interpretation atmospheric and hydrological loading are associated with IC1, IC2 and IC3. Since we do not have any clear evidence that IC4 is associated with hydrological or atmospheric loading, while studying the reduction of the standard deviation when removing the atmospheric and hydrological loading contribution, we decide to remove only the contribution of the first 3 ICs (in line 477 we say four, it was a mistake that we have now corrected).

**6. You state the ICs likely contain a larger component of power-law noise. When you are comparing the different filtering approaches (line 477-479) when you remove the modelled GFZ NTAL and HYDL are you also estimating a noise model for those series as well?**

In the following figures we show the Spectral Index (Fig. 1) and the white noise percentage (Fig. 2) of the IGb14-time series minus the displacements due to HYDL+NTAL. The white noise percentage of the resulting time series is smaller than what is obtained performing the

ICs filtering (Fig. 12b). We added this information in Sect. 5.1 and added Fig. 1 and Fig. 2 in the Supplementary material (Fig. S19).

[Figure]

Figure 1: Histograms of the spectral index in the filtered time-series. The filtering is done by subtracting the displacements due to HYDL+NTAL from IGb14-time series.

[Figure]

Figure 2: Histograms of the white noise percentage in the filtered time-series. The filtering is done by subtracting the displacements due to HYDL+NTAL from IGb14-time series.

**7. Lines 433-435 are a touch confusing. Are you assuming that removing the ICs completely removes all the annual signal? Could there not be other processes that have annual signals that are not captured by the ICs?**

We are assuming that the majority of the seasonal signals are captured by the ICs. As a result, we assume that there is no need to include annual and semi-annual terms in the time series modeling used to estimate the GNSS station vertical velocities.

It is worth noting that the inclusion of the annual and semi-annual terms in the time series modeling does not change much the estimation of the GNSS stations velocities, as you can see in the Fig. 3: the variation of the velocity estimates change less than 0.05 mm/yr for 479 of the 545 stations.

[Figure]

Figure 3: difference between GNSS vertical velocity estimation with and without considering annual and semi-annual terms in the time series modeling.

**8. Figure 6 might be improved by consistent axes.**
Ok, done.

**9. Maybe I missed it, but what precipitation data are you using (lines 557).**
The precipitation data we use are provided by the NASA Goddard Earth Sciences Data and Information Services Center (Huffman et al., 2019), they are daily with a spatial resolution of 0.1°. This information is reported in Section 3.2 (lines 252-253); we also added the link to the data in the "Code and data availability" section (line 702).